# The Fake Mirror Effect: Foreign Feedback Disrupts Self-Correction in Minimal Recurrent Networks

**Sungmoon Ong**  *softkorea@gmail.com*
*Independent Researcher*

Reviewed on OpenReview: *https://openreview.net/forum?id=cwENvGCLRv*

## Abstract

When a recurrent network iteratively refines its predictions, is the performance gain driven primarily by generic temporal integration, or does it depend on the specific geometry of the model's own prior output? To resolve this, we dissect a minimal 35-neuron recurrent network with controlled feedback interventions. Recurrent-specific gain is isolated by two contrasts that do not depend on the convergence-sensitive variable-noise residual: static repeated input, where any deterministic stateless aggregator yields zero gain by construction, and a 120k-parameter MNIST extension where the recurrent loop retains a small but reliable residual. Under variable noise a loop−ensemble residual is additionally present at low-to-moderate noise but is convergence-dependent (reported descriptively), with the no-feedback within-network ensemble matching or nominally exceeding the loop at high noise.

Testing feedback-source dependence directly, we replace self-generated feedback with structurally valid output from an independently trained clone. This consistently degrades performance relative to self-feedback at all tested scales. At the minimal scale, clone feedback drops accuracy below the no-feedback baseline—a scale-bounded *fake-mirror inversion* (robust under static input; marginal under variable noise). This absolute-harm penalty is scale-dependent in both regimes: under variable noise it transitions to a net-positive but still suboptimal signal at the larger tested widths and on MNIST, while under static input it attenuates toward near-neutral at the larger widths; coordinate-shuffled feedback by contrast is harmful at every tested scale.

Progressively stronger static aligners recover the clone-feedback loss essentially fully under static input but plateau near ∼85% under variable noise, indicating that static open-loop mapping does not fully restore closed-loop compatibility across the aligner capacities we evaluated. A relative condition-number analysis does not separate the evaluated feedback-source conditions. Together, these results characterize *feedback-geometry compatibility*: an empirical relation between a recurrent receiver and the geometry of its self-generated feedback that is not captured by generic temporal integration alone.

## 1 Introduction

### 1.1 The Inseparability Question

Chain-of-thought prompting, self-consistency decoding, and test-time compute scaling all assume that a model benefits from processing its own prior output—but would the output of an equally competent stranger work just as well? If a model's recurrent weights are co-adapted to its own output geometry, substituting another model's output should degrade performance even when that output is statistically valid. If, on the other hand, any well-formed feedback signal suffices, the benefit is generic rather than self-specific. The

---

*During the preparation of this manuscript, large language models were used to provide language refinement and structured feedback on the experimental plan. All scientific decisions, experiments, and claims are the author's responsibility.

question has practical relevance: methods relying on external test-time guidance—such as iterative external critics or verifier models—implicitly assume this signal is compatible with the model's internal processing, an assumption our results suggest deserves scrutiny at scale.

An analogous coupling exists in biological systems: prefrontal systems are implicated in metacognitive self-monitoring (Fleming & Dolan, 2012). Whether artificial neural networks exhibit a similar inseparability—where removing the self-referential pathway degrades the system—remains experimentally untested.

## 1.2   Why 35 Neurons

Our architecture sits at a critical intersection: it is *task-minimal* yet *fully transparent*. This scale breaks in both directions: smaller networks lack the capacity to reliably develop self-correction, while larger ones lose complete simulatability.

At 35 neurons (325 parameters), the system admits substantial causal dissection, placing this work squarely in the toy-model tradition of mechanistic interpretability (Olah et al., 2020; Elhage et al., 2022). The 50-parameter recurrent matrix can be arithmetically decomposed, hidden activations logged in their native 10-dimensional space, and causal interventions—such as wrong-trajectory substitutions and per-neuron knockouts—performed exactly; visual summaries are explicitly identified as projections of these activations.

We adopt the logic of Hodgkin & Huxley (1952)'s squid giant axon: the preparation's scale permits direct causal measurements that are infeasible in the opaque models where these phenomena are typically deployed. We do not claim biological relevance—the analogy is about methodology, not mechanism. What matters is whether the *phenomenon* observed here also appears at larger scales, not whether 35 neurons are themselves representative. We address this directly: scale verification shows that self-correction and the coordinate-shuffled-feedback penalty (formally condition C1, defined in §2.2) persist up to hidden width 245 and on MNIST at 120k parameters. The regime-specific absolute-harm component (clone feedback worse than no feedback; formally condition C2, defined in §2.2) does not persist uniformly beyond minimal scale (§ 3.9). This minimal-model framing is an operational premise rather than a scale claim: if feedback compatibility is a coherent constraint, its behavioral signature should be traceable as architecture and scale change. We therefore map where the signature appears, attenuates, or disappears across the 35-neuron-to-MNIST band, without committing to mechanism continuity across that band.

## 1.3   Prior Work

Our investigation lies at the intersection of three established literatures.

**Emergent self-reference.** Metacognitive capabilities can emerge spontaneously in recurrent architectures. Recent work shows tiny RNNs discover biological cognitive strategies (Li et al., 2025) and uncertainty monitoring (Ma et al., 2025) without an explicit metacognitive objective, while evolutionary simulations yield self-referential metamemory (Yamato et al., 2022). Theoretically, self-referential meta-learning has been proposed as a route to reducing explicit meta-optimization (Kirsch & Schmidhuber, 2022). Our system is self-referential in a weaker sense—output feedback rather than parameter self-modification—but this suffices to test whether the recurrent pathway develops model-specific co-adaptation.

**Iterative computation.** Recurrent networks naturally support temporal integration and iterative refinement. Two prominent architectures train models to dynamically determine their per-input computational depth via explicit halting mechanisms. ACT (Adaptive Computation Time; Graves, 2016) augments each timestep with a learned halting probability and accumulates outputs weighted by the remaining halting budget until cumulative halting probability reaches one. PonderNet (Banino et al., 2021) replaces this with a Bernoulli-like halting distribution and a regularization penalty toward a target average depth, yielding a probabilistic per-input compute budget. Both treat depth as a learned, input-dependent variable. More broadly, recurrent systems accumulate evidence from noisy observations over time (Gold & Shadlen, 2007), a trajectory-refinement process often analyzed through fixed-point attractor dynamics (Sussillo & Barak, 2013). These works establish that recurrent loops can effectively refine predictions.

**Solution degeneracy.** Concurrently, a robust body of work demonstrates that independently trained neural networks develop wildly divergent internal geometries. Despite identical architectures and tasks, neural dynamics exhibit massive solution degeneracy across random initializations (Maheswaranathan et al., 2019; Huang et al., 2025). To quantify and bridge these structural incompatibilities, techniques like model stitching (Bansal et al., 2021) employ learned affine transformations to map representations between frozen layers of different networks.

**Closed-loop co-adaptation.** The general principle that a controller must be compatible with the system it regulates is well established in control theory, and analogous distribution-shift problems arise in reinforcement learning when reward models are over-optimized beyond their calibration range (Gao et al., 2023). Our work can be understood as an empirical demonstration of this principle in the specific context of recurrent output feedback: the recurrent weights and the output geometry are jointly trained within a single network, and substituting a foreign output geometry into this loop creates a closed-loop mismatch analogous to controller-plant incompatibility. We use the term *feedback-geometry compatibility* (defined in §1.5) to refer to this empirical compatibility relation as observed in our minimal recurrent setting.

## 1.4 The Gap

These two facts—that recurrent loops enable iterative refinement, and that independently trained models develop incompatible representations—raise an untested question: *does this incompatibility causally degrade self-correction when one model's output is substituted into another's feedback loop?*

While generic distribution-shift theory predicts degradation when geometrically incompatible vectors enter a nonlinear closed loop, the observed empirical structure is finer-grained. Some forms of foreign feedback actively degrade performance below the no-feedback baseline, with the penalty bounded by perturbation type and scale. Specifically, our intervention set resolves this harm into two operational perturbation families with distinct robustness profiles: coordinate-shuffled feedback remains harmful at every tested scale, whereas absolute harm from clone-feedback is scale- and regime-dependent (§3.9). Beyond this scale-bounded harm pattern, static open-loop alignment recovers essentially all of the clone-feedback damage under static input but only $\sim 85\%$ under variable noise, where it plateaus despite a $3.7\times$ capacity increase. Finally, the variable-noise gain shows a positive loop−ensemble residual at low-to-moderate noise (matched by the ensemble at high noise); the recurrent-specific residual is further isolated by the static-input contrast and at MNIST scale. Together, these observations define the gap this paper studies: when feedback is structurally valid but geometrically foreign, its utility depends on whether it preserves the coordinate and geometric structure expected by the recurrent receiver.

## 1.5 Terminology and Scope

We use the following terms operationally: each refers to an architectural intervention or a measured quantity, distinct from language reasoning or general self-reference in large models.

**Closed-loop feedback (standard term).** The architectural property of feeding a network's output back into one of its inputs at a subsequent timestep. The recurrent baseline implements closed-loop feedback in this sense.

**Self-reference (this paper).** The specific case of closed-loop feedback in which the signal occupying the loop is the network's own prior output rather than an external signal or another network's output. Groups C1 (shuffled), C2 (clone), and the wrong-trajectory control preserve the architectural closed loop while replacing the self-feedback signal. Our interventions evaluate zero-shot substitution into a frozen recurrent receiver trained under self-feedback. This design leaves untested whether a target network could co-adapt to a clone's output under joint optimization; our target's receiver weights are trained only with its own output geometry. We therefore scope the reported compatibility relation to zero-shot substitution into a frozen receiver, rather than to all possible jointly trained feedback systems. We treat the resulting substitution penalty as a possible analogue for inference-time foreign-feedback injection, such as external critic or verifier signals.

**Correction gain (this paper).** The per-trial accuracy difference $\mathrm{acc}_{t=3} - \mathrm{acc}_{t=1}$, our operative measurement. We use integration controls to ask how much of this gain is reproducible by stateless temporal

aggregation, and reserve the term *closed-loop residual* for contrasts in which the recurrent baseline exceeds the matched stateless control or in which stateless gain is zero by construction (the static-input contrast).

**Evidence accumulation (standard term).** The reduction of predictive uncertainty by temporal integration of repeated noisy observations. A stateless integrator (e.g., a no-feedback ensemble that aggregates per-step outputs by log-product or probability-mean) performs evidence accumulation without self-reference. By definition, a deterministic stateless integrator on identical static inputs yields zero gain.

**Feedback-geometry compatibility (this paper, "fake-mirror effect" in §3.3).** The empirical dependence of recurrent improvement on whether the feedback signal preserves the coordinate and geometric structure expected by the recurrent receiver. Operationally, we observe (a) self-feedback yields positive correction gain, (b) coordinate disruption (C1) reverses that gain at every tested scale, and (c) clone feedback (C2) is worse than self-feedback across the tested settings and is also worse than no feedback in the primary minimal-scale setting, with scale/regime dependence characterized in §3.9. The "fake-mirror" name refers to the small-scale C2-vs-Group-A inversion.

**Where this fits (chain-of-thought analogy).** The motivating phenomena introduced in §1.1—chain-of-thought prompting, self-consistency decoding, test-time compute scaling—all use intermediate model-generated outputs across multiple inference steps. The closed-loop architecture studied here is a minimal architectural analogue of one structural ingredient of that family. We offer the following one-to-one structural mapping:

| What maps | Limits of analogy |
| --- | --- |
| Multi-step reasoning chain $\leftrightarrow$ $t{=}1 \to t{=}2 \to t{=}3$ unroll | Token-level autoregressive generation $\leftrightarrow$ 5-class fixed-step classification |
| Self-consistency / self-verification $\leftrightarrow$ self-feedback (Baseline) | Semantic reasoning correctness $\leftrightarrow$ categorical accuracy only |
| Foreign intermediate output $\leftrightarrow$ clone feedback (C2) | Prompted-by-template reasoning $\leftrightarrow$ architectural closed loop with no prompt |
| Reasoning gain from extra steps $\leftrightarrow$ correction gain $\text{acc}_{t=3} - \text{acc}_{t=1}$ | Scale-emergent reasoning in LLMs $\leftrightarrow$ untested beyond 35-neuron and MNIST-120k settings |

The mapping is structural: both involve the iterative use of self-generated information.

## 1.6 Contributions

1. **Trained recurrent self-reference produces a closed-loop gain that stateless integration is mathematically precluded from producing under identical inputs.** Group A (no recurrence) yields exactly zero gain in FP64 across all 20 seeds, whereas the recurrent Baseline obtains +0.036 under static input (a deterministic stateless aggregator on identical inputs must yield exactly zero gain by construction). The +0.036 magnitude itself depends on the deferred-commitment loss regime ($w_1{<}0.3$; App. D). Under variable noise, the loop's gain is +0.189; a loop$-$ensemble residual of +0.040 at low-to-moderate noise ($\sigma{\leq}0.5$, $p = 0.007$; not significant at high noise, $\sigma{\geq}0.7$) is additionally present but convergence-sensitive (Appendix J); the recurrent-specific claim rests on the static-input contrast and the MNIST-scale residual. The closed-loop residual that emerges in the static-input contrast persists at MNIST scale, where the recurrent Baseline maintains a +0.0064 advantage over the same ensemble (paired Wilcoxon $p = 1.34 \times 10^{-5}$). Single-pass terminal-step ablation (Group A) eliminates the gain in all settings. The necessity claim concerns the trained receiver as a learned system: we do not test whether an untrained-$W_{\text{rec}}$ receiver (identity-initialized) could replicate the residual, which would refine the necessity to architectural state-passing rather than learned dynamics.

2. **Feedback utility depends on geometric source, with scale-dependent magnitude.** Replacing a model's self-feedback with a clone's output degrades performance relative to self-feedback at every tested scale; at minimal scale, clone feedback also degrades performance below the no-

feedback baseline (a scale-bounded fake-mirror inversion: robust under static input, and marginal under variable noise—the variable-noise (VN) A-vs-C2 contrast is Holm $p$=0.025). This absolute-harm component attenuates with scale: under variable noise it transitions to a net-positive but still suboptimal signal at the larger tested widths and on MNIST, while under static input it attenuates toward near-neutral (mean slightly negative; the per-width CI overlaps zero at $w{\geq}45$). The minimal-scale absolute-harm inversion is reported at the primary ($\tau$=2.0, $w_1$=0, $T$=3) training configuration; Baseline self-correction emergence is verified across 40 VN configurations of $(\tau, w_1, w_2)$ (§3.8), and the static C2 inversion is confirmed in a targeted spot-check at $\tau \in \{1.5, 3.0\}$ and $w_1$=0.1 (§4.4 item 6).

3. **Foreign-feedback interpolation produces smooth degradation.** Interpolation between self-generated and foreign feedback reveals a smooth, broadly monotonic degradation of correction gain, with no sharp collapse threshold. Magnitude alone does not explain the harm: per-trial L2-rescaling of clone feedback to match self-feedback's L2 norm leaves the gain significantly negative, and the rescaling does not appreciably rotate the post-tanh contribution direction.

4. **Static open-loop alignment saturates below full recovery under variable noise.** Progressively stronger static alignment models (30 to 17,925 parameters) trained to minimize MSE against target self-feedback on independently collected open-loop trajectories recover most-to-essentially-all of the static-input C2 drop (affine 76%, the MLP family 93–99%), but under variable noise saturate at ∼85% recovery, with the residual not closing despite a 3.7× capacity increase from MLP-medium to MLP-large. Appendix G shows that a task-supervised closed-loop adapter can bridge the gap, but only by using a saturation-heavy regime outside the receiver's native feedback distribution; this makes the ceiling specific to label-free open-loop alignment rather than absolute.

5. **Local conditioning is insufficient to dissociate the feedback-source conditions.** A relative condition number analysis at the feedback receiver does not pairwise dissociate any of the four tested feedback conditions; the behavioral ordering in §3.5 is more directly characterized by the trajectory-level analyses than by the single-step Jacobian condition number.

## 2 Methods

### 2.1 Architecture, Task, and Training

We constructed a recurrent MLP: Input(10) $\rightarrow$ H1(10) $\rightarrow$ H2(10) $\rightarrow$ Output(5), totaling 35 neurons (325 parameters). Hidden layers use ReLU; the output is linear. At each timestep $t \in \{1, 2, 3\}$, the network computes (in row-vector convention, matching the NumPy implementation):

$$f_t = \tanh(y_{t-1}/\tau) \quad (t > 1; \; f_1 = \mathbf{0}) \tag{1}$$

$$h_t^{(1)} = \mathrm{ReLU}(x_t \, W_{ih_1} + f_t \, W_{\mathrm{rec}} + b_{h_1}) \tag{2}$$

$$h_t^{(2)} = \mathrm{ReLU}\left(h_t^{(1)} \, W_{h_1 h_2} + b_{h_2}\right) \tag{3}$$

$$y_t = h_t^{(2)} \, W_{h_2 o} + b_{\mathrm{out}} \tag{4}$$

with $W_{ih_1} \in \mathbb{R}^{10 \times 10}$, $W_{h_1 h_2} \in \mathbb{R}^{10 \times 10}$, $W_{h_2 o} \in \mathbb{R}^{10 \times 5}$, and the single recurrent matrix $W_{\mathrm{rec}} \in \mathbb{R}^{5 \times 10}$, which carries the feedback signal $f_t$ from the previous output back into H1. The tanh nonlinearity acts on the *feedback path* (Eq. 1), not on the output: $y_t$ itself is unactivated. Temperature $\tau = 2.0$ was chosen empirically to keep typical feedback magnitudes bounded and reduce tanh saturation in the operating range. The "recurrent weight ablation" condition (Group A, §2.2) sets only $W_{\mathrm{rec}}$ to zero post-training; all feedforward weights $\{W_{ih_1}, W_{h_1 h_2}, W_{h_2 o}\}$ and biases are unchanged.

The primary 35-neuron implementation is in pure NumPy (no deep learning frameworks) so that every computation is directly inspectable; numerical gradient checking verified correctness (details in Appendix A). An independent PyTorch reimplementation of the same architecture is used as a cross-validation control in §3.6 and Appendix A.6, and the MNIST extension uses PyTorch separately. Figure 1 summarizes the architecture and the feedback loop.

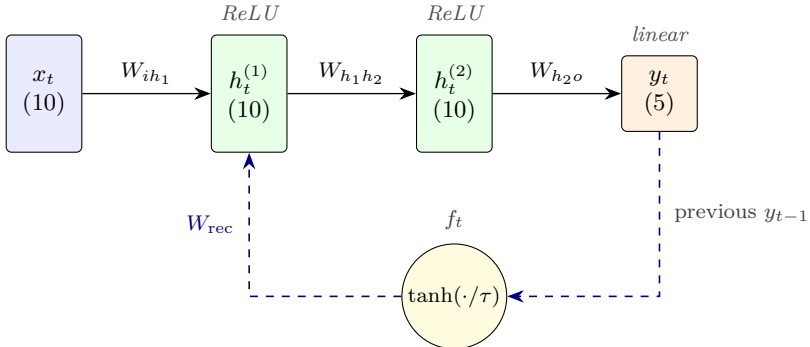

Figure 1: **Architecture of the 35-neuron recurrent MLP and feedback loop.** At each timestep $t$, input $x_t$ propagates through two ReLU hidden layers ($h_t^{(1)}$, $h_t^{(2)}$) to a linear output $y_t$ (Eqs. 1–4). The dashed path injects the previous output $y_{t-1}$ back into $h_t^{(1)}$ via $f_t = \tanh(y_{t-1}/\tau)$ and the single recurrent matrix $W_{\text{rec}}$. The four feedback conditions (§2.2) modify only the dashed path: **Baseline** uses self-feedback ($f_t$ from the model's own previous output); **Group A** sets $W_{\text{rec}}=0$, severing the dashed path; **C1** (shuffled) independently re-permutes the coordinates of the feedback vector $f_t$ at each feedback timestep; **C2** (clone) substitutes $y'_{t-1}$ from an independently trained donor model.

The task is a 5-class prototype classification problem with deliberately overlapping adjacent-class features (prototype amplitudes in the Setup box below). This class structure was chosen to ensure non-trivial within-trial correctability—neither so easy that step 1 suffices nor so hard that step 3 fails—rather than to mirror any particular real-world dataset. In the primary setting (variable noise, § 2.3), independently sampled Gaussian noise is added per timestep, enabling temporal evidence accumulation. A complementary static setting provides the *identical* noise-corrupted pattern at all $T = 3$ timesteps; because the input is then constant, the recurrent loop cannot accumulate independent input samples—it can only carry information about past outputs, isolating self-correction from input-side temporal integration.

---

**Setup at a glance: prototypes and one corrected trial**

Each class $k \in \{0, \ldots, 4\}$ is characterized by a *prototype* vector $p_k \in \mathbb{R}^{10}$ that serves as its clean reference pattern. Dimensions $2k$ and $2k+1$ are strongly activated (amplitude 1.0), and one flanking dimension on each adjacent-class side—dimensions $(2k - 1) \bmod 10$ and $(2k + 2) \bmod 10$—is partially activated (amplitude 0.3); the adjacency is circular, so classes 0 and 4 also share leakage. The 0.3 leakage encodes *inter-class ambiguity*: a single-shot noisy input is often consistent with two prototypes simultaneously.

| $k$ | $p_k[0]$ | $p_k[1]$ | $p_k[2]$ | $p_k[3]$ | $p_k[4]$ | $p_k[5]$ | $p_k[6]$ | $p_k[7]$ | $p_k[8]$ | $p_k[9]$ |
|---|---|---|---|---|---|---|---|---|---|---|
| 0 | 1.0 | 1.0 | 0.3 | 0 | 0 | 0 | 0 | 0 | 0 | 0.3 |
| 1 | 0 | 0.3 | 1.0 | 1.0 | 0.3 | 0 | 0 | 0 | 0 | 0 |
| 2 | 0 | 0 | 0 | 0.3 | 1.0 | 1.0 | 0.3 | 0 | 0 | 0 |
| 3 | 0 | 0 | 0 | 0 | 0 | 0.3 | 1.0 | 1.0 | 0.3 | 0 |
| 4 | 0.3 | 0 | 0 | 0 | 0 | 0 | 0 | 0.3 | 1.0 | 1.0 |

This design is the minimal setting in which iterative refinement is non-trivial: without inter-class overlap, the $t=1$ output would be far less ambiguous and self-correction would be less diagnostic. Under variable noise (§2.3), later timesteps can combine fresh per-timestep noisy evidence with output feedback; under static input, the same architecture isolates output-feedback refinement without fresh evidence. Five classes with adjacent-class overlap provide a compact configuration in which the system retains this ambiguity while remaining directly analyzable at 35 neurons (325 parameters).

*One corrected trial* (Baseline self-feedback, seed 0, trial 164 of the mechanistic-analysis cohort of §3.5; true class = 1). Input $x_t = p_1 + \varepsilon$ with $\varepsilon \sim \mathcal{N}(0, 0.5^2 I)$ held identical across the three timesteps (static input). The model's logits over the five classes and its argmax prediction at $t = 1, 2, 3$:

| $t$ | $y_0$ | $y_1$ | $y_2$ | $y_3$ | $y_4$ | argmax |
|---|---|---|---|---|---|---|
| 1 | $-3.72$ | $+1.07$ | $+1.25$ | $-0.70$ | $-0.93$ | 2 (incorrect) |
| 2 | $-3.12$ | $+3.25$ | $+2.36$ | $-1.84$ | $-4.06$ | 1 (corrected) |
| 3 | $-2.60$ | $+3.27$ | $+1.89$ | $-2.40$ | $-4.33$ | 1 (confirmed) |

The initial guess (class 2) reflects the model's $t=1$ read of the noisy input alone, before any feedback signal becomes available. Self-feedback at $t=2$ reverses the prediction to the true class 1 (the target logit $y_1$ rises from $+1.07$ to $+3.25$), and $t=3$ confirms it. This is one realization of a *Corrected* trial in the trial-outcome taxonomy of §3.5 (Figure 4c); aggregated over 20 model seeds, 10.9% of trials follow this Corrected pattern under self-feedback versus 7.7% under clone feedback ($p_{\text{Holm}} < 0.01$).

**Training protocol.** Models were trained via full-batch gradient descent (lr = 0.01, 1000 epochs, 200 train / 200 test samples) using 3-step BPTT. We used vanilla full-batch SGD over adaptive optimizers for inspectability of every gradient step in the pure-NumPy implementation; empirically, this 325-parameter setup with 200 samples converged reliably under the chosen learning rate, so adaptive optimizers were not needed for the reported results. The 1000-epoch budget is the operating point at which correction gain has largely converged (terminal accuracy continues to improve under longer budgets, so the reported gains are conservative); convergence and overfitting diagnostics are reported in Appendix I.

**Time-weighted objective.** The time-weighted cross-entropy loss is $L = \frac{w_1 \cdot L(t=1) + w_2 \cdot L(t=2) + 1.0 \cdot L(t=3)}{w_1 + w_2 + 1.0}$ with $w = (0.0, 0.2, 1.0)$; normalizing by $\sum_i w_i$ (here 1.2) decouples the effective learning rate from the weight schedule. Setting $w_1 = 0.0$ removes the penalty on the initial output, creating optimization pressure to use the feedback loop for iterative refinement. Because all timesteps share the same weights, the unpenalized $t=1$ output still benefits from gradient flow through $t=2$ and $t=3$, achieving non-trivial accuracy even without direct loss. Correction gain ($\text{acc}_{t3} - \text{acc}_{t1}$) thus measures the improvement from an unpenalized intermediate guess to a fully penalized final output. This explicit time-weighting is sufficient but not necessary: under variable noise, self-correction emerges even with uniform weighting ($w_1 = w_2 = 1.0$; § 3.8).

## 2.2 Experimental Conditions

Against a baseline, ten evaluation conditions isolate the causal role of self-referential feedback. We organize the baseline and these conditions into four functional families (detailed architectures and hyperparameters for Groups D$'$, D$''$, and C2-mlp are provided in Appendix C):

**1. Baseline and structural ablations.** Standard controls testing structural dependence on the recurrent pathway:

- **Baseline:** The fully trained recurrent model, evaluated normally.

- **Group A (Recurrent ablation):** The single recurrent weight matrix $W_{\text{rec}}$ is set to zero post-training (equivalently, the recurrent contribution is omitted at evaluation, as in the scale and MNIST implementations); all feedforward weights and biases are unchanged. Preserves feedforward capacity but severs the feedback signal.

- **Group B1 (Random ablation):** 50 randomly zeroed weights (30 repeats/model) to control for generic parameter loss.

- **Group B2 (Structured ablation):** The entire h2-to-output weight matrix zeroed (output bias retained) as a catastrophic structural sanity check.

**2. Feedforward controls.** Stateless networks trained from scratch, verifying that capacity and depth cannot substitute for recurrence:

- **Group D:** Standard feedforward network.

- **Group D$'$ (Parameter-matched):** Adds a skip connection to exactly match the recurrent model's parameter count.

- **Group D$''$ (Compute-matched):** A deep 6-layer network matching the computational depth and multiply-add operations of the Baseline's 3-timestep unroll.

**3. Feedback identity manipulation (the "fake mirror").** These novel conditions preserve the recurrent architecture but alter the feedback signal to test for model-specific co-adaptation:

- **Group C1 (Shuffled feedback):** The feedback vector is independently re-permuted at each timestep (seeded per evaluation repeat). This maintains the exact marginal statistics (distribution, norm) while destroying coordinate identity.

- **Group C2 (Clone feedback):** Self-feedback is replaced with the output of an independently trained "clone" (identical architecture and procedure, independent data; target seeds 0–19, donor seeds 100–119). Because the clone's output is a mathematically valid, in-distribution classification vector, this condition is designed to probe trajectory-level geometric incompatibility while controlling for gross output validity (statistical limits of this isolation are addressed in §3.7). By using an equal-capacity clone we aim to isolate the geometric compatibility cost from any capability gap; we confirm this capability match directly (§3.3): donors reach terminal-step accuracy statistically indistinguishable from their paired targets. Whether a higher-capacity donor's feedback could override this penalty is left untested.

**4. Alignment controls (quantifying C2 mismatch).** To decompose the C2 performance drop, we fitted progressively stronger transformations from donor logits to target logits on held-out self-feedback calibration data. Aligned donor outputs were injected during evaluation:

- **Group C2-aligned (Affine):** A fitted linear transform (30 parameters) to disentangle linear representation misalignment.

- **Group C2-mlp (Non-linear family):** Fitted MLPs of increasing capacity ($5 \rightarrow 16 \rightarrow 5$ at 181 parameters up to $5 \rightarrow 128 \rightarrow 128 \rightarrow 5$ at 17,925 parameters) to test whether stronger non-linear mappings can close the residual gap.

*Together, these quantify how much of the "fake-mirror" degradation is reducible representation mismatch versus persistent incompatibility that static open-loop alignment cannot resolve (details in Appendix C).*

## 2.3 Variable-Noise Task

To enable temporal evidence accumulation, we introduce a variable-noise (VN) variant: $x_t = \text{prototype}_k + \varepsilon_t$, with independently sampled $\varepsilon_t \sim \mathcal{N}(0, \sigma^2 I)$ per timestep. Target models and 20 donor models (seeds 100–119) were retrained under variable noise using the baseline hyperparameters ($w_1$=0.0, $w_2$=0.2, $\tau$=2.0, 1000 epochs). During evaluation, paired target and donor networks received identical per-timestep noise sequences, ensuring output differences strictly reflect model identity rather than input variation. A noise-level sweep ($\sigma \in \{0.0, 0.1, \ldots, 1.0\}$) probed the dependence of correction gain on task difficulty. We use independent Gaussian noise as a minimal structural proxy for iterative refinement under uncertainty; this differs from resolving structured semantic ambiguity (e.g., chain-of-thought reasoning), and we make no claim that temporal denoising and semantic search share underlying geometry.

## 2.4 Interpolation and Mechanistic Analyses

The following exploratory analyses used the static-input cohort ($N$=20, noise = 0.5):

**Feedback interpolation.** We swept $\alpha \in \{0.0, 0.1, \ldots, 1.0\}$ in $\text{feedback}_t = \alpha \cdot y_{t-1}^{\text{self}} + (1-\alpha) \cdot y_{t-1}^{\text{other}}$, testing zero, shuffle, and clone contaminations.

**Mechanistic tracking.** (a) Internal activations were logged to classify trials by correction outcome. (b) Decision-space dynamics were visualized in the 2D plane defined by the target logit $y_{\text{true}}$ and the maximum distractor logit $\max_{c \neq \text{true}} y_c$ (Figure 4); the original Hidden-Layer-1 PCA projection used in earlier drafts is preserved as a supplementary diagnostic in Appendix B. (c) We computed the per-trial cosine divergence between self and clone recurrent ($W_{\text{rec}}$) contributions.

**Neuron importance.** To quantify single-neuron contributions without baseline-collapse confounds, we performed decoupled knockouts (seed = 0). For H1, *intelligence importance* was measured by zeroing a neuron's feedforward inputs (its $W_{ih_1}$ column and bias; observing $\Delta\text{acc}_{t1}$), and *correction importance* by zeroing its recurrent inputs (its $W_{\text{rec}}$ column; observing $\Delta\text{gain}$). Hidden Layer 2 neurons underwent full knockouts.

## 2.5 Supplementary Experiments

Seven additional experiments probe specificity and generalization:

(a) *Wrong-trajectory control:* to disentangle foreign-model identity from state-conditional mismatch, each model's feedback was replaced with its own output from a different trial (matched on predicted class and $t$=1 correctness; when multiple candidates matched, one was selected uniformly at random using a fixed seed), continuously substituted at every $t \geq 2$.

(b) *Cross-pairing:* 400 donor–target pairings ($20 \times 20$) under VN to assess donor variance.

(c) *Divergence null baselines:* isotropic random vectors (norm-matched per trial to the self-feedback contribution; upper bound) and same-class resampled self-feedback (lower bound) to contextualize clone divergence.

(d) *Training dynamics:* correction gain tracked at 16 epoch checkpoints ($N$=20) with fully trained donor.

(e) *Timestep extension:* evaluation at $T \in \{3, 5, 7, 10, 15, 20\}$ without retraining.

(f) *Scale verification:* four hidden widths (10, 20, 45, 245) under static and VN, 20 models each.

(g) *MNIST extension:* replicated on MNIST ($\sim$120k parameters) to test generalizability (architecture details in Appendix C).

## 2.6 Statistical Design and Robustness

Primary experiments used $N$=20 independently initialized models (seeds 0–19); exploratory analyses (§ 2.4) used the same cohort. The inferential unit is the model, not the repeat. The primary measure is **correction gain** ($\text{acc}_{t3} - \text{acc}_{t1}$).

Significance was assessed with exact Wilcoxon signed-rank tests with Holm–Bonferroni correction. Pre-specified families: static $m$=4 (Baseline vs. {A, B1, C1, C2}), VN $m$=3 (Baseline vs. {A, C1, C2}), secondary $m$=2 (C1-vs-A, C2-vs-A). 95% bootstrap CIs (10,000 resamples, resampling model-level paired differences as a unit to preserve pairing structure) are reported as descriptive summaries. Groups B2, D, D′, D″ were excluded from the Holm families *a priori* as structural controls.

The inferential unit is the trained model: each independently seeded model is its own control across conditions (e.g., Baseline and Group A are evaluated on the same target network with only its $W_{\text{rec}}$ set to zero post-training). Paired comparison subtracts model-level variation, isolating the within-model condition effect—the standard repeated-measures pattern, applied here at the model level. We use the Wilcoxon signed-rank test because it does not assume normality of paired differences (a small-$N$ diagnostic we cannot reliably make at $N$=20); we use the *exact* (combinatorial) variant rather than the large-sample normal approximation because at $N$=20 the discrete null distribution is finite and tractable. Tests are two-sided for conservatism, even when the expected direction was specified by the intervention design. Holm step-down within each pre-specified

family controls the family-wise error rate at the same level as Bonferroni while being less conservative. Bootstrap CIs accompany the $p$-values as descriptive complements, not as a separate inferential procedure.

**Claim hierarchy and multiplicity.** We control the family-wise error rate *within* each pre-specified family rather than through a single global correction, because the families address distinct, structurally separate questions organized hierarchically; pooling them into one Bonferroni family would be inappropriate for largely independent questions and inconsistent with the family structure stated at each result. The confirmatory families are the primary-gain contrasts (static $m$=4, VN $m$=3; §3.1), the secondary foreign-feedback-vs-no-feedback contrasts (C1-vs-A and C2-vs-A, corrected separately per regime, $m$=2; §3.3), the closed-loop alignment family (Appendix G, $m$=4), and the local-conditioning $\kappa$ contrasts (§3.7, $m$=6, robust per-seed median as the primary statistic). We report $p$-values uncorrected with the family stated at each result and give the Holm-adjusted value where it bears on a conclusion.

**Descriptive analyses and anchoring.** Quantities we report descriptively are not entered into any confirmatory family and are flagged as such at point of use: the integration-control $\sigma$-sweep residuals (per-$\sigma$ and the pooled $\sigma \leq 0.5/\sigma \geq 0.7$ bands; §3.6), which are pre-declared but convergence-dependent (Appendix J) and therefore reported descriptively, with unadjusted exploratory $p$-values—the low-band residual is +0.040 (raw $p$=0.007) under the converged protocol but null under the submitted protocol; the tail-sensitive mean-$\kappa$ contrasts (the median is primary); the feedback interpolation (§2.4); the seed-level dose–response null (§3.5); and the wrong-trajectory magnitude. The central claims do not rest on the convergence-sensitive §3.6 low-noise sign flip: they rest on the static-input gain—positive at every protocol endpoint in Appendix J, and exactly zero for any deterministic stateless aggregator by construction—and on the MNIST-scale loop−ensemble residual, evaluated in a separate scale experiment.

Robustness was assessed via hyperparameter sweeps: 80 static hyperparameter configurations ($w_1 \times w_2 \times \tau$ grid, 10 models each) and 40 VN hyperparameter configurations (10 models each; 400 model-runs total), including truly uniform time weighting ($w_1 = w_2 = 1.0$). Grid details are provided in Appendix D.

## 3 Results

### 3.1 The Baseline: Recurrent Feedback Produces Terminal-Step Gain

Under variable noise, the recurrent Baseline improved from an initial $t$=1 accuracy of 0.713 to a final $t$=3 accuracy of 0.902. This yields a substantial correction gain of +**0.189 ± 0.053** ($N$=20, 95% CI [+0.166, +0.212]). Crucially, this gain is not an artifact of the unpenalized initial step in the time-weighted loss; under truly uniform time weighting ($w_1 = w_2 = 1.0$), all 40 of 40 models reproduce a positive gain (§ 3.8). Under static input (constant noise across timesteps), the Baseline achieves a smaller but significant correction gain of +0.036 ± 0.051 (95% CI: [+0.014, +0.058], $N$=20). By providing identical inputs across timesteps, this static setting isolates self-correction from input-side temporal integration, establishing a clean foundation for the mechanistic analyses in § 3.5. As decomposed in §3.6, the loop−ensemble residual is positive at low-to-moderate noise ($\sigma \leq 0.5$, raw $p$=0.007, reported descriptively) and not significant at higher noise; the recurrent-specific component is further isolated by the static-input contrast and by the MNIST-scale residual reported in §3.9.

**Removing recurrence eliminates the single-pass terminal-step gain.** Ablating the recurrent weights (Group A) entirely eliminated the terminal-step improvement under both settings (VN: +0.013±0.038, Static: 0.000), leaving $t$=1 and $t$=3 accuracies effectively identical. The Group A result measures single-pass terminal accuracy with the recurrent contribution disabled; the separate integration-control analysis in §3.6 quantifies what changes when stateless outputs are aggregated post hoc. The Group A contrast therefore isolates the recurrent pathway under single-pass terminal evaluation specifically, not stateless temporal integration: the Baseline's terminal-step accuracy gain is produced via the feedback loop, not by raw single-pass refinement. Noise-level dependence is characterized in §3.4.

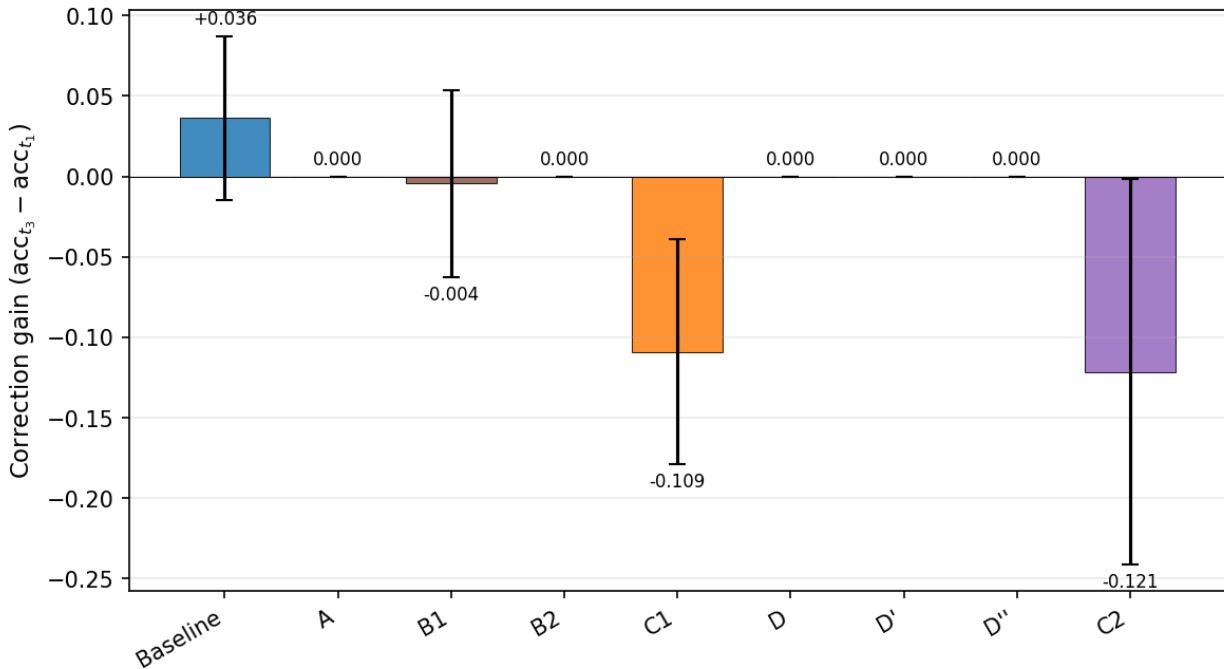

Figure 2: Correction gain across experimental conditions (static input, noise $= 0.5$, $N{=}20$ seeds; error bars show $\pm 1$ SD). Baseline achieves positive gain; Group A (recurrent ablation) yields exactly zero; C1 and C2 produce negative gains worse than no feedback. Feedforward controls (D, D$'$, D$''$) show zero gain regardless of capacity.

### 3.2 Feedforward and Ablation Controls

**Ablation controls confirm structural specificity.** Group A's initial $t{=}1$ accuracy (0.748, static input) matches the Baseline perfectly, confirming that recurrent ablation selectively neutralizes the correction mechanism without impairing feedforward recognition. Conversely, random parameter loss (B1) and severing the hidden-to-output pathway (B2) collapse $t{=}1$ accuracy to 0.554 and a chance-level 0.205, respectively. This demonstrates that generic network damage destroys basic pattern recognition rather than isolating the self-correction loop.

**Feedforward controls rule out capacity and depth.** Three stateless architectures—D (standard), D$'$ (parameter-matched), and D$''$ (compute-matched 6-layer)—yielded zero correction gain under static single-pass evaluation. Neither additional parameters nor computational depth produced terminal-step correction gain without recurrent feedback (Figure 2).

By design, under static input the parameter-matched D$'$ model achieves a higher single-pass accuracy (0.858 at $t{=}1$) than the recurrent Baseline's refined accuracy (0.784 at $t{=}3$). We deliberately operate in a capacity-constrained regime to force the network to utilize temporal recursion, isolating the iterative refinement mechanism from raw parameter scaling.

### 3.3 The Fake Mirror Effect

**At minimal scale, foreign feedback can be worse than nothing.** In the primary 35-neuron setting, shuffled feedback (C1) and static clone feedback (C2) drive the gain *below* the no-feedback baseline; the variable-noise clone-vs-A contrast is significant but marginal (Holm $p{=}0.025$). While removing recurrence yields a structurally guaranteed zero gain, foreign feedback can actively drive the gain negative rather than merely failing to help.

Replacing a model's self-generated feedback with non-veridical signals produced negative mean correction gain—performance at $t=3$ dropped below the initial $t=1$ prediction on average:

- **Shuffled feedback (Group C1):** gain $= -0.102 \pm 0.092$ (VN) and $-0.109 \pm 0.065$ (Static CI: $[-0.137, -0.081]$). Permuting the feedback vector destroys coordinate identity while preserving its exact marginal statistics.

- **Clone feedback (Group C2):** gain $= -0.040 \pm 0.101$ (VN) and $-0.122 \pm 0.120$ (Static CI: $[-0.175, -0.070]$). Substituting the mathematically valid, in-distribution output of an independently trained clone also drove the correction gain negative, below the no-feedback baseline in both settings (most strongly under static input; the scale/regime dependence of this C2 harm is characterized in §3.9).

Critically, both substitutions performed worse than severing the feedback loop entirely (Group A, no feedback, gain $= 0.000$ static $/ +0.013$ VN)—strongly so under static input and for C1 under variable noise, and significantly but marginally for the VN C2-vs-A contrast (Holm $p=0.025$; C2's own VN gain CI includes zero). **Statistical summary:** All primary comparisons (Baseline vs. A, C1, C2) are highly significant (Holm–Bonferroni $p < 0.00002$ for VN), with bootstrap 95% CIs for the Baseline−condition paired differences excluding zero (Table 1). Secondary comparisons (C1-vs-A and C2-vs-A, separately corrected family $m=2$) confirm that foreign feedback is significantly worse than no feedback in both regimes (Static: A-vs-C1 Holm $p = 3.8 \times 10^{-6}$, A-vs-C2 Holm $p = 4 \times 10^{-4}$; VN: A-vs-C1 Holm $p = 3.8 \times 10^{-5}$, A-vs-C2 Holm $p = 0.025$). Processing structurally valid but foreign feedback misdirects the network more severely than receiving no feedback at all.

We refer to this pattern as *feedback-geometry compatibility* (defined in §1.5): the recurrent loop is tuned to the geometry of its own outputs, so even in-distribution foreign feedback can be disruptive. We examine the boundary between this observed compatibility relation and standard closed-loop control-theory accounts in § 4.2.

As a descriptive cross-pairing check (400 donor–target pairings under variable noise; these reuse 20 targets $\times$ 20 donors and are not 400 independent systems, and all pairings are evaluated on a single shared test draw, so we report them descriptively rather than with a pairing-level CI), the mean C2 gain is $-0.037$ (range $[-0.355, +0.210]$; per-target mean SD $= 0.052$), with 264/400 pairings below the no-feedback Group A gain ($+0.013$)—consistent with the C2 degradation generalizing across initializations.

A donor-capability control confirms the clone is capability-matched rather than merely weaker (primary static setting, $N=20$, each model evaluated in self-feedback on the targets' test inputs): donor terminal-step accuracy is statistically indistinguishable from target (0.769 vs 0.784, paired Wilcoxon $p=0.53$; t1 differs by a non-significant 0.036, $p=0.054$), so the C2 drop is not a capability deficit. Target and donor agree on 64% of t1 argmax predictions on identical inputs, so the C2 perturbation in principle mixes continuous-geometry divergence with class-level disagreement (cf. §4.4 limitation 5). Stratifying the C2 evaluation by this agreement label tests this decomposition: on the 64% agreement subset, self-feedback still outperforms clone feedback by $+0.144$ (per-seed mean Baseline−C2 gain) and on the 36% disagreement subset by $+0.180$; the across-subset difference is not significant (paired Wilcoxon $p=0.22$). The agreement-stratified result preserves most of the relative-advantage gap; we do not detect an additional class-level perceptual contribution in this stratification.

### 3.4 Continuous Dynamics: Noise and Feedback Interpolation

**Feedback interpolation.** To test whether the learned compatibility relation is binary or smoothly varying, we swept the self-feedback fraction $\alpha$ from 1.0 (pure self) to 0.0 (pure contamination). Correction gain degrades smoothly and broadly monotonically across substitution types (Figure 3), revealing a soft compatibility relation with no sharp collapse threshold. The zero-crossing occurs at $\alpha \approx 0.4$ for shuffled and $\alpha \approx 0.55$ for clone contamination, indicating that structured misdirection requires more veridical self-signal to overcome than coordinate-disrupted feedback. Recovery is largely complete by $\alpha \approx 0.7$ for zero and

Table 1: Primary results ($N$=20). Correction gain ($\mathrm{acc}_{t3} - \mathrm{acc}_{t1}$) with bootstrap 95% CIs and exact Wilcoxon signed-rank $p$-values (uncorrected; all remain significant after Holm–Bonferroni correction with static $m$=4, VN $m$=3). Paired differences are Baseline minus each condition.

| Condition | Gain (mean ± SD) | 95% CI | BL−X [CI] | $p$ |
|---|---|---|---|---|
| *Panel A: Static Input* | | | | |
| Baseline | $+0.036 \pm 0.051$ | $[+0.014, +0.058]$ | — | — |
| Group A | $+0.000 \pm 0.000$ | $[+0.000, +0.000]$ | $+0.036\ [+0.014, +0.058]$ | $5.4 \times 10^{-3}$ |
| Group B1 | $-0.005 \pm 0.023$ | $[-0.014, +0.005]$ | $+0.041\ [+0.022, +0.060]$ | $1.0 \times 10^{-3}$ |
| Group C1 | $-0.109 \pm 0.065$ | $[-0.137, -0.081]$ | $+0.145\ [+0.123, +0.167]$ | $1.9 \times 10^{-6}$ |
| Group C2 | $-0.122 \pm 0.120$ | $[-0.175, -0.070]$ | $+0.158\ [+0.116, +0.203]$ | $1.9 \times 10^{-6}$ |
| *Panel B: Variable Noise* | | | | |
| Baseline | $+0.189 \pm 0.053$ | $[+0.166, +0.212]$ | — | — |
| Group A | $+0.013 \pm 0.038$ | $[-0.003, +0.030]$ | $+0.175\ [+0.156, +0.197]$ | $1.9 \times 10^{-6}$ |
| Group C1 | $-0.102 \pm 0.092$ | $[-0.143, -0.062]$ | $+0.291\ [+0.256, +0.323]$ | $1.9 \times 10^{-6}$ |
| Group C2 | $-0.040 \pm 0.101$ | $[-0.084, +0.004]$ | $+0.229\ [+0.191, +0.268]$ | $1.9 \times 10^{-6}$ |

shuffled contamination, while clone feedback continues to recover gradually toward self-feedback levels up to $\alpha$=1.0 (full tabular data in Appendix E).

**Noise dependence.** Correction gain exhibits an inverted-U relationship with task difficulty, forming a broad peak across intermediate noise levels ($\sigma = 0.5$–$0.6$; gain $= +0.189$ at the primary $\sigma = 0.5$ operating point)[1] and declining at both extremes. At low noise, single-pass feedforward processing suffices; at high noise, the signal is unrecoverable. Self-correction thus provides the greatest advantage in an intermediate regime where initial predictions are frequently incorrect but the underlying signal remains recoverable through iterative refinement.

### 3.5 Mechanistic Dissection and Alignment Decomposition

Having shown that foreign feedback degrades performance smoothly rather than catastrophically (§ 3.4), we next ask what exactly is mismatched when that feedback enters the recurrent loop.

**Decision-space behavior.** Trials were classified by correction outcome: *corrected* (wrong at $t$=1, correct at $t$=3; 10.9%), *stable correct* (68.1%), *stable incorrect* (15.4%), and *over-corrected* (correct at $t$=1, wrong at $t$=3; 5.7%).[2] In output-logit decision space (Figure 4a,b), two coordinates suffice for the discrete argmax outcome: the target logit $y_{\mathrm{true}}$ and the maximum distractor logit $\max_{c \neq \mathrm{true}} y_c$.[3] Self-feedback shifts trial mass upward across the decision boundary $y$=$x$ (mass above boundary: $+5.2$ pp from $t$=1 to $t$=3), whereas clone feedback shifts it in the opposite direction ($-9.1$ pp). Across BL/C2 trials, the maximum-distractor identity ($\arg\max_{c \neq \mathrm{true}} y_c$) matches between $t$=1 and $t$=3 in $68.6\% \pm 7.2\%$ of trials per seed (range 47.5–81.2%). A residual $\sim$31% axis-non-stationary contribution therefore remains; restricting to the axis-stable subset ($\sim$65–72% of trials per condition) preserves the shifts with the same sign and order of magnitude (Baseline $+3.5$ pp, C2 $-6.9$ pp), so axis relabeling does not drive the displayed shift. This decision-space shift is mirrored at the trial-outcome level (Figure 4c): clone feedback reduces corrected-trial mass and increases over-correction (Corrected $-3.2$ pp, Stable incorrect $+3.1$ pp, Over-corrected $+11.1$ pp; all $p_{\mathrm{Holm}} < 0.01$, paired Wilcoxon over $N$=20 seeds). The upstream geometric mismatch associated with these decision-space differences is examined numerically below (cosine divergence) and in §3.7 (Jacobian conditioning).

**Wrong-trajectory control** ($N$=20). To test whether this misdirection stems from foreign-model identity or merely state-conditional mismatch, we replaced each model's feedback with its own output from a different,

---

[1]The extended 11-level sweep peaks at $\sigma = 0.5$ ($+0.18$); the $\sigma = 0.6$ sample mean is essentially tied ($+0.177$; paired difference $-0.003$, paired Wilcoxon $p = 0.65$).

[2]The decision-space breakdown, mass shifts, and Figure 4 are computed on the mechanistic-analysis cohort (test seed 1000+$s$, static input; Baseline gain $+0.052$ on this draw), distinct from the primary held-out cohort (seed $+500$; $+0.036$) of §3.1; cf. the independent-draw note in Appendix E. The net $+5.2$ pp mass shift equals corrected $-$ over-corrected on this cohort.

[3]Off-manifold dynamics in the orthogonal distractor dimensions are not visible in this projection.

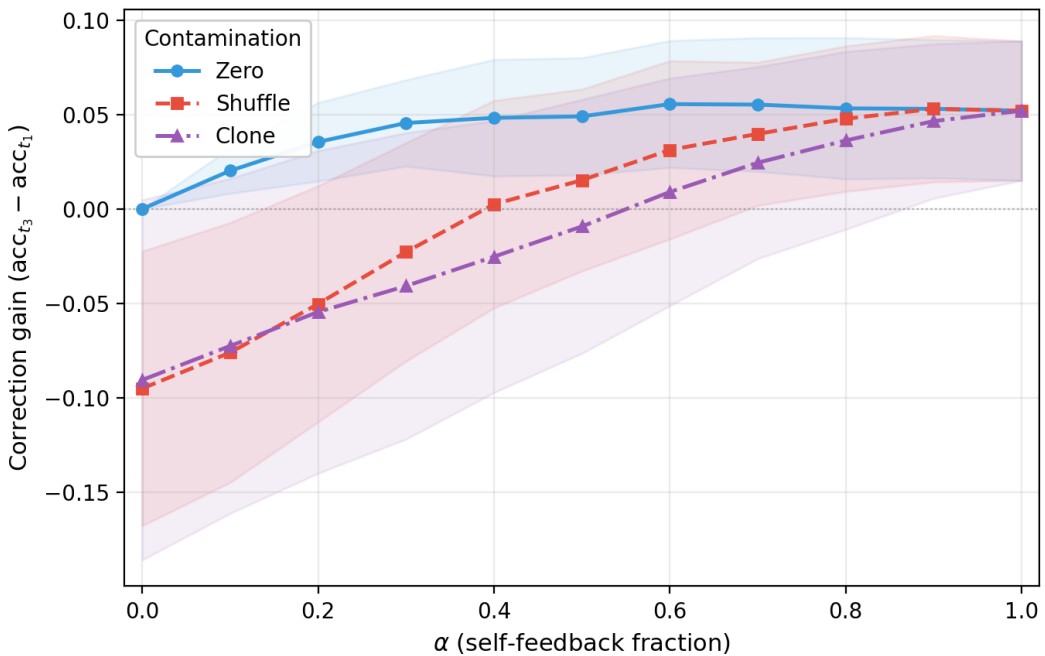

Figure 3: Feedback interpolation: correction gain as a function of self-feedback fraction $\alpha$, where $\text{feedback}_t = \alpha \cdot y_{t-1}^{\text{self}} + (1 - \alpha) \cdot y_{t-1}^{\text{other}}$. Three contamination signals ($y^{\text{other}}$) are tested: zero feedback, shuffled feedback, and clone feedback. Shaded bands show $\pm 1$ SD across $N{=}20$ seeds. Degradation is smooth and broadly monotonic with no sharp threshold; zero-crossings differ by contamination type ($\alpha \approx 0.4$ for shuffle, $\approx 0.55$ for clone).

class-matched trial. Same-model wrong-trial feedback preserved positive gain (static: $+0.057$, VN: $+0.108$; the static value exceeds the primary Baseline partly because the substituted trial injects an uncorrelated noise draw—an ensembling benefit, §4.4 limitation 4), whereas clone feedback degraded it (static: $-0.091$, VN: $-0.049$).[4] Same-model wrong-trial feedback is therefore less disruptive than clone feedback (the local-sensitivity geometry of these conditions is analyzed in §3.7).

**Clone feedback geometry.** Although two models may produce similar discrete predictions, their continuous logit vectors can drive recurrent weights in conflicting directions. Across all test trials, the cosine divergence—defined as $1-\cos(\theta)$ (range 0 for identical, 1 for orthogonal, 2 for antiparallel)—between $f_t^{\text{self}} W_{\text{rec}}$ and $f_t^{\text{clone}} W_{\text{rec}}$ averages 0.635 (Figure 5), falling between two reference baselines: same-class resampled self-feedback (0.342, where the feedback is computed from the same model's output on a different class-matched trial) and isotropic random vectors (1.000). Clone feedback thus occupies an intermediate position: more aligned than noise but substantially less aligned than self-feedback.

**Seed-level association.** Across the 20 model seeds, per-seed divergence does not significantly predict per-seed C2 degradation (Spearman $\rho = -0.17$, $p = 0.46$, $N{=}20$).[5] Aggregate geometric mismatch thus remains consistent with the group-level effect, but does not exhibit a monotone seed-level dose-response: higher divergence in a given model does not reliably correspond to larger C2 harm in that same model. This null test evaluates aggregate model-level traits; whether per-trial divergence predicts per-trial harm (which might reveal a localized dose-response masked by seed-level aggregation) remains an open question. The cosine and L2-matching controls bound but do not eliminate the reading that clone feedback acts merely as an out-of-distribution input to the frozen receiver at inference (cf. the zero-shot scoping in §1.5). A target

---

[4]Clone gains here are evaluated on the matched-trial subset and may differ slightly from the full-sample C2 values in § 3.3.

[5]At $N{=}20$ the Spearman test has 80% power for $|\rho| \gtrsim 0.44$, so the observed $\rho{=}-0.17$ is reported as failure to find evidence, not as evidence of independence.

Table 2: Alignment saturation under variable noise ($N$=20). Recovery is the per-seed mean of (gain − $\text{gain}_{\text{C2}}$)/($\text{gain}_{\text{BL}}$ − $\text{gain}_{\text{C2}}$)—the fraction of the Baseline–C2 gap closed, computed per seed and then averaged over the 20 seeds. Because per-seed gaps vary, this seed-averaged ratio is lower than the ∼90% obtained from the ratio of the column-mean gains shown.

| Alignment | Params | VN Gain | Recovery |
|---|---|---|---|
| C2-raw (no alignment) | 0 | −0.040 | 0% |
| Affine | 30 | +0.133 | 68% |
| MLP-small (5-16-5) | 181 | +0.156 | 80% |
| MLP-medium (5-64-64-5) | 4,869 | +0.166 | 85% |
| MLP-large (5-128-128-5) | 17,925 | +0.166 | 85% |
| Baseline (self-feedback) | — | +0.189 | 100% |

trained from scratch with donor feedback in the loop would provide the cleanest disambiguation, an approach we leave to future work.

**Alignment decomposition ($N$=20).** To quantify how much of this geometric incompatibility is simply static representation mismatch, we fitted learned transformations mapping donor to target logits. Recovering the Baseline-to-C2 gap yielded a distinct three-part decomposition:

- **Linear alignment is insufficient:** An affine transform (30 params) recovers the majority but not all of the lost gain (static: 76%, VN: 68%).

- **Learned alignment recovers most of the gain:** A learned MLP recovers essentially all of the static clone harm (∼93–99% across 181–17,925 parameters; differences across MLP sizes are within seed noise) but saturates near 85% under variable noise.

- **A persistent residual remains under variable noise:** Increasing alignment capacity from 4,869 to 17,925 parameters yields no further variable-noise recovery (saturating at ∼85%), and the aligned gain remains significantly below Baseline (BL vs. MLP-large $p = 0.0047$ under VN; targeted residual test, not part of the pre-specified Holm-corrected families). All alignment models were trained with vanilla SGD using a 20% sample-level validation split with best-epoch model selection to control overfitting; the pattern persists under this regularized regime.

Under variable noise this saturation is a key finding (Table 2): progressively stronger static alignment shows diminishing returns and a persistent ∼15% gap that added capacity does not close, whereas under static input a learned aligner recovers essentially all of the harm. The variable-noise residual reflects a limitation of static open-loop alignment for the closed-loop feedback task, rather than merely insufficient alignment capacity within the tested model family (full alignment family details in Appendix C).

Appendix G reports a complementary in-loop, task-supervised closed-loop BPTT alignment ablation as the matched in-loop counterpart to the static, open-loop, label-free family characterized here. That ablation performs above Baseline and the donor contributes beyond an $x_t$-only control (the label-leakage diagnostic is not triggered, donor marginal +0.03–0.055), but most of the gain is already attained by the $x_t$-only control, and it bridges the gap by driving the receiver into a heavily saturated feedback regime that the static-MSE family does not access by construction (operational definition in Appendix G); we return to the interpretive consequences in §4.4 limitation 3 and §5 conclusion item 3.

**Structural and single-neuron diversity (exploratory).** Supporting this model-specific interpretation, $W_{\text{rec}}$ sign consistency across 20 independently trained models averaged 0.563. This intermediate consistency suggests that multiple distinct weight configurations can implement effective self-correction, rather than a single universal algorithm. (Ablation controls confirm this recurrent dependence is not a ReLU dead-zone artifact: recurrent ablation leaves every hidden unit active rather than driving any into a dead, all-zero regime.) Furthermore, decoupled single-neuron knockouts (§2.4—for each Hidden Layer 1 neuron, ablating *only* its feedforward input column in $W_{ih_1}$ (with its bias) to obtain *intelligence importance* $\Delta\text{acc}_{t1}$, or

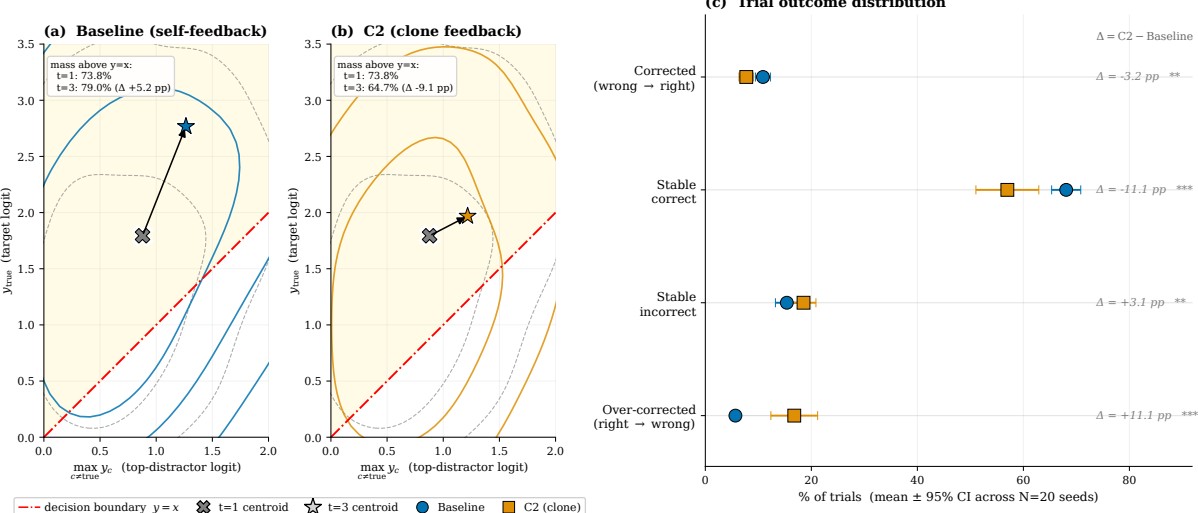

Figure 4: Behavioral consequence in output-logit decision space. **(a, b)** Joint distribution of the target logit $y_{\text{true}}$ versus the maximum distractor logit $\max_{c \neq \text{true}} y_c$ at $t{=}1$ (gray dashed contours) and $t{=}3$ (panel-colored solid contours; blue = Baseline, orange = C2 clone) over $\sim$4,000 test trials pooled across 20 model seeds. Yellow shading marks the correct-prediction half-plane ($y_{\text{true}} > \max_{c \neq \text{true}} y_c$); red dash-dotted line is the decision boundary $y{=}x$. Centroid markers ($\times$ at $t{=}1$, $\star$ at $t{=}3$) and black arrows show migration. Boxed annotations report the percentage of trials above the boundary. Under self-feedback (a) the distribution shifts upward ($+5.2$ pp correct-zone mass); under clone feedback (b) it shifts downward ($-9.1$ pp). **(c)** Distribution of trial outcomes across four categories: *Corrected* (wrong at $t{=}1$, correct at $t{=}3$), *Stable correct*, *Stable incorrect*, *Over-corrected* (correct at $t{=}1$, wrong at $t{=}3$). Clone feedback shifts mass from the recovery categories (Corrected, Stable correct) to the failure categories (Stable incorrect, Over-corrected). Markers report seed-mean $\pm$ 95% CI ($N{=}20$ model seeds); thin lightgray segments connect the Baseline and C2 means within each outcome category. $\Delta = \text{C2} - \text{Baseline}$ in percentage points; stars denote Holm-corrected paired Wilcoxon significance ($^{**}\, p_{\text{Holm}} < 0.01$, $^{***}\, p_{\text{Holm}} < 0.001$).

*only* its recurrent input column in $W_{\text{rec}}$ to obtain *correction importance* $\Delta$gain; seed = 0) revealed diverse internal wiring. Categorizing each H1 neuron by the joint pattern of the two importance metrics yielded a partition into intelligence-specialized (high $\Delta\text{acc}_{t1}$, low $\Delta$gain), correction-specialized (high $\Delta$gain, low $\Delta\text{acc}_{t1}$), and shared (high in both) functional groups. Not all recurrent channels are constructive; ablating h1_9's recurrent input actively *improved* correction gain (correction importance $= -0.015$, i.e., the gain rose by 0.015 after ablation), underscoring the idiosyncratic nature of the learned compatibility relation. Full single-neuron heatmaps and Layer 2 analyses are deferred to Appendix F.

## 3.6 Integration Control: Decomposing Variable-Noise Gain

**Within-network ensembling accounts for part, not all, of the variable-noise gain.** To test whether the unrolled gain is reproduced by a variance-reducing ensemble, we compare the recurrent Baseline against a no-loop control (E1): the same trained weights with the recurrent contribution set to zero—identical to the network's own first-pass (no-feedback) computation, so E1 is three in-distribution forward passes on the same inputs, combined post hoc as a log-product or probability-mean ensemble ($N{=}20$). The earlier feedforward controls (D, D′, D″; §3.2) and the recurrence-ablation control (Group A; §3.1) measure *single-pass terminal accuracy*; E1 measures *post-hoc temporal aggregation* of the same network's per-step outputs. The paired loop$-$ensemble residual is $+0.034$ to $+0.044$ across $\sigma \in \{0.1, 0.3, 0.5\}$ (per-$\sigma$ exact Wilcoxon $p = 0.047, 0.004, 0.015$; pooled low-band aggregate $+0.040$, unadjusted $p = 0.007$, reported descriptively)

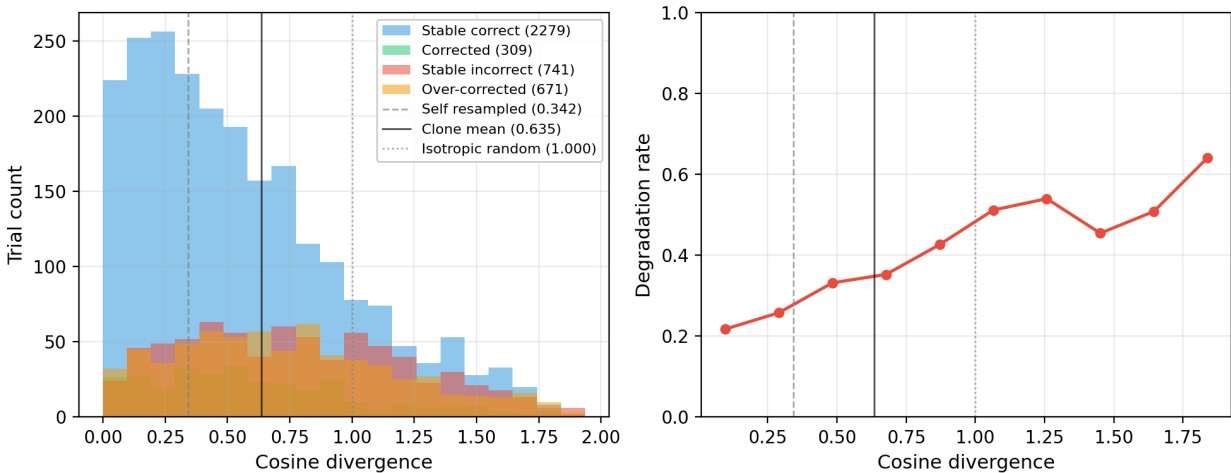

Figure 5: Cosine divergence between self and clone $W_{rec}$ contributions, computed per-trial at $t{=}2$ across 4,000 C2 trials pooled across 20 model seeds. *Left panel:* distribution by C2 trial outcome (stacked histogram), with vertical lines marking the three reference baselines (resampled self-feedback 0.342, clone mean 0.635, isotropic random vectors 1.000). Clone divergence falls between the two reference distributions, confirming partial but insufficient geometric alignment. *Right panel:* binned degradation rate (fraction of trials classified as stable-incorrect or over-corrected) as a function of cosine divergence; trend rises with divergence but does not separate cleanly from the soft-modulation regime.

and is not significant for $\sigma \geq 0.7$ ($p = 0.73, 0.09$); equivalently, E1 recovers a noise-dependent fraction of the gain ($\approx$18%, 59%, 79% at $\sigma$=0.1–0.5, rising to $\geq$100% at $\sigma \geq 0.7$). At $\sigma$=1.0 E1 is nominally higher than the loop ($+0.149$ vs $+0.131$): feeding noisy outputs back through the loop compounds errors across timesteps, which the independent ensemble avoids. This regime structure is consistent with a variance-reduction component, supplied by within-network ensembling, that grows with input noise. The conclusion is unchanged under probability-averaging ensembles (low-band $p = 0.004$), and the comparison replicates at MNIST scale (§3.9) and in an independent PyTorch reimplementation (Appendix A.6). We therefore treat the recurrent benefit as partially but not wholly reducible to within-network ensembling: at high noise the gain is consistent with variance reduction, while at low-to-moderate noise ($\sigma \leq 0.5$) a positive loop$-$ensemble residual remains.

**E1 on static input: a formal control.** By definition, a deterministic stateless aggregator on identical inputs produces identical per-timestep logits at every step, so any aggregator's argmax equals the single-pass argmax and the gain is mathematically zero. We empirically verify this property in our implementation: the per-seed E1 static gain is exactly 0.000000 across all 20 seeds in FP64. The Baseline's static gain of $+0.036$ (§3.1) therefore reflects iterative refinement that no stateless aggregator can reproduce on identical inputs.

**E1 across the noise sweep.** Under variable noise, E1 produces nontrivial gain because the per-timestep inputs are independently noisy and the aggregator combines independent observations. We evaluate E1 (log-product aggregation) alongside the recurrent Baseline at five noise levels ($\sigma \in \{0.1, 0.3, 0.5, 0.7, 1.0\}$, $N$=20, exploratory comparisons reported with unadjusted exact paired Wilcoxon two-sided $p$-values):

| $\sigma$ | Baseline gain | E1 gain (log-product) | E1/Baseline | Paired Wilcoxon $p$ |
|---|---|---|---|---|
| 0.10 | $+0.042$ | $+0.008$ | 0.18 | 0.047 |
| 0.30 | $+0.108$ | $+0.064$ | 0.59 | 0.004 |
| **0.50** | **$+0.189$** | **$+0.148$** | **0.79** | **0.015** |
| 0.70 | $+0.173$ | $+0.172$ | 1.00 | 0.726 |
| 1.00 | $+0.131$ | $+0.149$ | 1.14 | 0.095 |

**Cross-implementation check (PyTorch).** The integration-control comparison replicates in an independent PyTorch reimplementation: the low-noise loop−ensemble residual is positive (low-band +0.045, exploratory unadjusted $p = 0.002$; $\sigma$=0.5 residual +0.051, $p = 0.003$; Appendix A.6), confirming the effect is not specific to the NumPy implementation.

**Scale verification (MNIST, ~120k parameters).** On a 10-class MNIST extension with the same recurrent MLP architecture, the recurrent Baseline retains a significant residual advantage over the within-network E1 ensemble: the recurrent Baseline gain is +0.0188, the within-network E1 log-product gain is +0.0123, and the per-seed paired residual is +0.0064 (bootstrap 95% CI [+0.0046, +0.0081]; paired Wilcoxon $p = 1.34 \times 10^{-5}$). This corroborates the minimal-scale low-noise residual at a larger scale and a different task: the recurrent loop contributes a measurable closed-loop residual that the no-feedback ensemble does not capture.

### 3.7 Local Conditioning and Magnitude-Matched Perturbations

We complement the foregoing intervention experiments with a quantitative local analysis at the feedback receiver. For each trained Baseline model and each feedback condition (self, clone, wrong-trajectory, norm-matched clone), we compute the Jacobian of the H1 pre-activation contribution from the recurrent path with respect to the previous-output vector $y$:

$$J(y) = (W_{\text{rec}}^{\top}/\tau) \cdot \text{diag}(\text{sech}^2(y/\tau))$$

evaluated at the actually-injected feedback for each condition. We report the relative condition number $\kappa = \sigma_{\max}(J)/\sigma_{\min}(J)$ per (seed, condition) and pairwise paired Wilcoxon tests across conditions. For the local-sensitivity analysis, the wrong-trajectory condition uses the same-class resampled self-feedback protocol used for the cosine baseline, rather than the stricter predicted-class and $t$=1 correctness matching used in the behavioral wrong-trajectory control above.

**Local conditioning does not dissociate the feedback conditions.** For each condition, we compute the condition number $\kappa = \sigma_{\max}/\sigma_{\min}$ of the recurrent feedback Jacobian, evaluated along the feedback states actually injected by that condition ($N$=20 seeds). Because $\kappa$ is a strictly positive, heavy-tailed ratio—and saturated tanh units have $\tanh'(y/\tau) \to 0$ terms that can shrink $\sigma_{\min}$—we summarize typical local conditioning by the per-state median rather than by the tail-sensitive arithmetic mean. Typical conditioning is similar across conditions (median $\kappa$: self 13.1, clone 10.1, wrong-trajectory 12.5, norm-matched clone 14.0); by contrast, the per-seed mean $\kappa$ is treated only as a tail diagnostic and can reach several thousand on individual seeds. Using the robust per-seed median statistic, none of the six pairwise contrasts is significant even before Holm correction (minimum raw $p = 0.729$; all Holm-adjusted $p = 1.000$). The only nominal difference under the tail-sensitive mean statistic is self vs. wrong-trajectory (raw $p = 0.019$, Holm $p = 0.115$), but it vanishes under the robust median statistic ($p = 0.956$) and runs opposite to the behavioral ordering: the better-performing self condition has higher, not lower, tail-sensitive conditioning. Because a large $\kappa$ reflects a small $\sigma_{\min}$ (a near-singular feedback direction) rather than amplified forward gain, we treat $\kappa$ only as a local conditioning diagnostic; it does not provide a robust, behavior-aligned pairwise dissociation, and the explanatory weight remains with the trajectory-level analyses in §3.5.

**Magnitude-matched clone.** A natural concern is that the clone-feedback harm reflects only L2-magnitude differences in the pre-tanh logits. We tested a per-trial norm-matched clone control in which the clone's pre-tanh logits are rescaled to match the self-logits' L2 norm before entering the recurrent gate, evaluated under static input where the clone harm is strong (C2 = $-0.122$). The norm-matched clone gain remains strongly negative ($-0.103$, $p = 0.001$ vs zero) and is not significantly different from the un-matched static C2 ($\Delta = +0.018$, $p = 0.44$). The cosine divergence on $\tanh(y/\tau) \cdot W_{\text{rec}}$ of the norm-matched feedback (0.649) is essentially identical to the raw clone's (0.654; the same self-vs-clone divergence as the 0.635 of §3.5, recomputed here on this control's matched-trial static cohort rather than the independent static test draw used in §3.5), so L2-rescaling pre-tanh logits does not appreciably rotate the post-tanh contribution direction. Magnitude alone therefore does not explain the negative C2 gain; the harm persists at matched L2 norm.

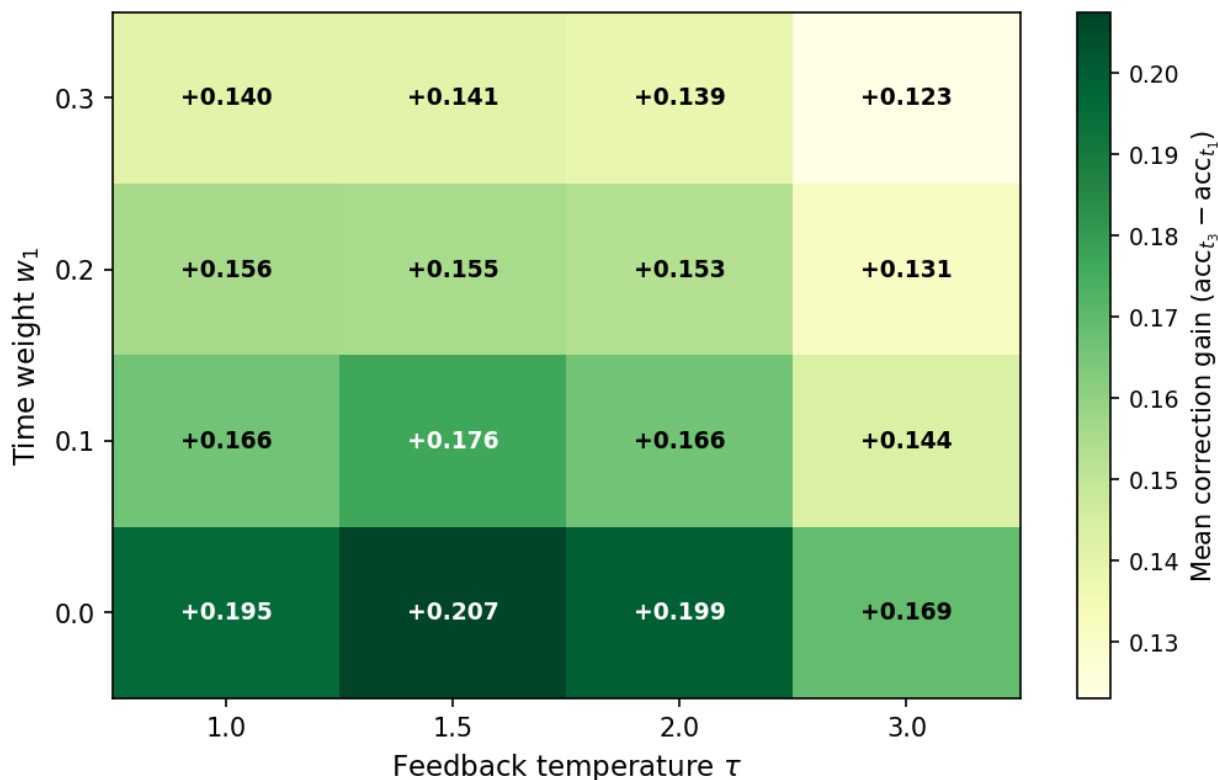

Figure 6: VN hyperparameter sweep (targeted grid): correction gain across $w_1$ and $\tau$ configurations ($w_2$=0.2 fixed, 10 seeds each). Self-correction emerges in 100% of these 16 configurations. An extended grid (24 additional configurations with $w_1 \in \{0.5, 0.7, 1.0\}$ and $w_2 \in \{0.2, 1.0\}$; Appendix D) also shows 100% emergence, yielding 40/40 total.

**Angular dimension as a remaining gap.** Norm-matching closes the L2-magnitude attack surface but does not close the angular-distance attack surface: clone feedback differs from self at cosine divergence 0.635 (§3.5), a same-class resampled-self proxy differs at 0.342, and the exact behavioral wrong-trajectory cohort sits lower still at $0.215 \pm 0.051$ (§4.4 limitation 5; `wrong_traj_cosine.csv`). Because the wrong-trajectory cohort also carries the independent-noise ensembling confound discussed in §4.4, whether the clone-vs-wrong-trajectory dissociation reflects source identity or angular distance remains open within the present interventional set. A parametric angular sweep at fixed L2 norm would be required to separate the two.

### 3.8 Robustness and Temporal Stability

**Robustness and spontaneous emergence.** Self-correction is not an artifact of a specific loss weighting. Under variable noise (VN), the phenomenon is highly robust (emerging in 100% of tested configurations; Figure 6) and arises spontaneously even under **truly uniform time weighting** ($w_1 = w_2 = 1.0$, all 40 models positive across four tau configurations of 10 seeds each). The explicitly time-weighted loss is therefore sufficient but not necessary: iterative refinement emerges naturally from the combination of a recurrent architecture and per-timestep noise variation. (Under static input, emergence was more limited at 59%; full grid details in Appendix D.)

**Training dynamics.** The compatibility relation forms gradually over $\sim$200 epochs ($N$=20). Vulnerability to shuffled feedback (C1) develops in parallel with self-correction. Clone feedback (C2) harm emerges later

($\sim$epoch 100). A caveat: because the donor is evaluated at its final trained state, this delayed C2 emergence may partly reflect progressive target-donor geometric divergence rather than purely intrinsic differentiation.

**Timestep extension.** Evaluating beyond the $T{=}3$ training horizon (up to $T{=}20$, no retraining) confirms stable convergence. Baseline terminal-step accuracy (at $t{=}T$) plateaus safely (static: $0.801 \to 0.775$; VN: $0.899 \to 0.895$ from $T{=}3$ to $T{=}20$), whereas non-veridical feedback (C1, C2) compounds slightly rather than self-correcting at higher $T$.

### 3.9   Scale Verification and MNIST Extension

**Scale verification.** To test whether the findings are specific to the 35-neuron setting, we evaluated hidden widths from 10 to 245 ($\sim$325 to 65k parameters). Self-correction and the C1 (shuffled-feedback) coordinate-disruption penalty persisted across all tested scales, the per-width C1$-$A difference excluding zero at every width (Figure 7; Table 9). The strict C2 baseline inversion (clone feedback worse than no feedback; §1.5) does *not* persist uniformly across scales: under variable noise, C2 transitions from absolute harm at minimal scale to a net-positive but still suboptimal signal at the larger tested widths ($w{\geq}45$); under static input, C2 attenuates toward near-neutral—the mean stays slightly negative, but the per-width C2$-$A CI no longer excludes zero at $w{\geq}45$ (Table 9), so the strict static inversion is statistically reliable only at $w \in \{10, 20\}$ (full per-width data in Appendix H). This pattern aligns with independently trained representations converging functionally at larger scales, rendering foreign feedback more geometrically compatible (Huh et al., 2024).

**MNIST extension ($\sim$120k parameters).** We applied the same recurrent architecture (Input(784) $\to$ H1(128) $\to$ H2(128) $\to$ Output(10) with output$\to$hidden feedback; 119,562 parameters, PyTorch) to a 10-class MNIST classification task. At each timestep $t \in \{1, 2, 3\}$, the model receives an independently noised view of the same training image ($x_t = \text{image} + \varepsilon_t$, $\varepsilon_t \sim \mathcal{N}(0, \sigma^2 I)$ per pixel with $\sigma = 0.5$)—i.e., the variable-noise regime of §2.3 transferred to MNIST. 20 target models and 20 donors trained for 30 epochs with mini-batch SGD; the four feedback conditions (Baseline / Group A / C1 / C2) are evaluated identically to the 35-neuron setting. Full configuration including determinism settings is in Appendix C.

On this task ($N{=}20$), self-correction replicates (Baseline gain $= +0.019 \pm 0.005$, all 20 models positive, $p < 0.0001$ vs. Group A) and the C1 coordinate-disruption penalty replicates (C1 gain $= -0.021 \pm 0.007$, every seed negative). The absolute-harm component (C2 worse than no feedback), however, does *not* replicate at MNIST scale: clone feedback (C2) yields a strictly positive correction gain ($+0.010 \pm 0.005$), indicating that foreign feedback at this scale is functionally utilized rather than actively harmful. A relative geometric advantage for self-feedback nonetheless remains, as clone feedback provides significantly less benefit than self-feedback (Baseline $-$ C2 $= +0.009$, paired Wilcoxon $p < 10^{-4}$; per-seed gains are available in the supplementary CSV files listed in Appendix H). The C1/C2 dissociation—coordinate-level disruption robust at every tested scale, geometric-source mismatch scale-dependent—clarifies which component of the original observation is more robust beyond minimal scale.

**Training dynamics.** Tracking MNIST training revealed an **inverted C2 trajectory**: C2 clone feedback is negative at epoch 1 but recovers to a positive gain by epoch 3, while C1 remains negative throughout. This pattern echoes representation divergence—foreign feedback may be more disruptive early in training when models are geometrically idiosyncratic—though a floor effect (low baseline accuracy at early epochs limits observable harm) and scale mismatch between the fully-trained donor and immature target may also contribute. The C1/C2 dissociation across epochs nonetheless remains descriptively informative, but we do not treat it as a clean causal isolate of representation divergence.

## 4   Discussion

### 4.1   Induction Depends on Conditions, Not Merely Scale

Self-correction developed in a network of 35 neurons—orders of magnitude smaller than systems where iterative refinement is typically studied. Under static input, temporal freedom ($w_1 < 0.3$) and gradient flow through the feedback pathway were important for reliable induction; static gain diminishes as $w_1$ increases and approaches zero at $w_1 = 0.3$, suggesting that static self-correction partly reflects deferred computation

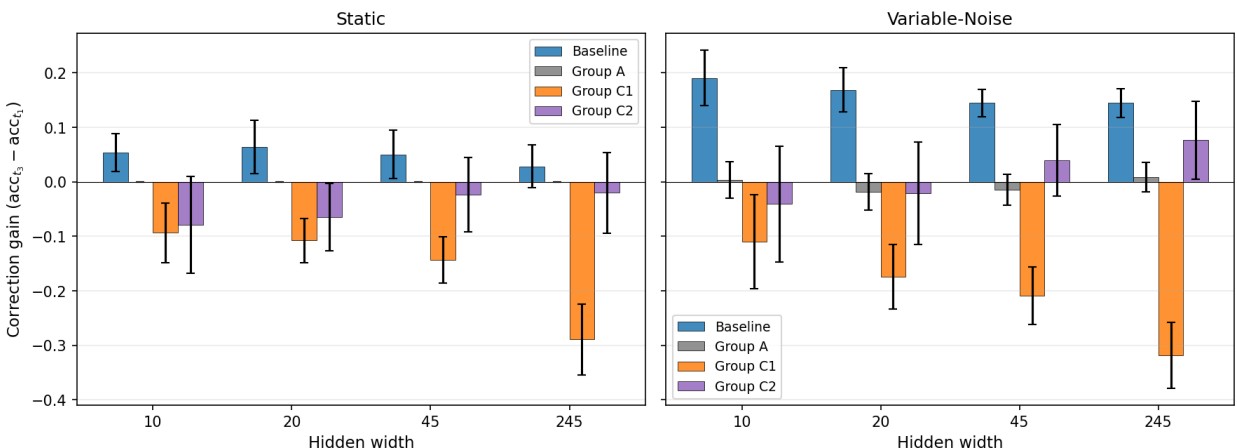

Figure 7: Scale verification across hidden widths 10–245 ($N$=20 seeds per condition; error bars show $\pm$1 SD). Self-correction (Baseline) and the C1 coordinate-disruption penalty persist at all tested scales. C2 (clone feedback) shows a scale-dependent transition: reliably negative at small scales ($w \in \{10, 20\}$), near-neutral at larger static widths (mean slightly negative but the per-width CI overlaps zero for $w{\geq}45$; Table 9), and net-positive but still suboptimal under variable noise.

enabled by reduced early-timestep loss pressure. Under variable noise, however, self-correction emerged even under uniform time weighting ($w_1 = w_2 = 1.0$, all 40 models positive; § 3.8), demonstrating that the phenomenon arises spontaneously from the combination of recurrent architecture and per-timestep noise variation, without requiring any special loss design. Under VN, Group A's terminal-step gain $\approx 0$ provides the empirical single-pass stateless baseline under the tested i.i.d. inputs. The informative result is then twofold: (i) the Baseline's variable-noise gain of $+0.189$, for which § 3.6 reports a positive loop$-$ensemble residual at low-to-moderate noise ($\sigma{\leq}0.5$) and no significant difference at high noise, and (ii) the Baseline's static gain of $+0.036$, a quantity that is exactly zero for any deterministic stateless aggregator.

One might argue that VN self-correction is simply Bayesian evidence accumulation. At $\sigma{\leq}0.5$ the loop$-$ensemble residual is $+0.040$ (pooled low-band, paired Wilcoxon $p = 0.007$, reported descriptively); at $\sigma{\geq}0.7$ the difference is not significant (the ensemble is nominally higher at $\sigma{=}1.0$). Single-pass stateless evaluation (Group A) shows terminal-step gain $\approx 0$ under these i.i.d. inputs—as do the feedforward controls (D, D$'$, D$''$) under static input—so the recurrent loop is necessary for the trained network's single terminal output to exploit the per-timestep observations (post-hoc stateless aggregation separately accounts for a noise-dependent fraction of the VN gain). The closed-loop residual isolated under recurrent self-reference is identified by the static-input gain ($+0.036$) and the MNIST-scale residual over the same ensemble ($+0.0064$); the $\sigma{\leq}0.5$ loop$-$ensemble residual is consistent with this but convergence-dependent (Appendix J) and not load-bearing.

## 4.2 Two Layers of Feedback-Geometry Compatibility: Coordinate vs. Geometric

Within our intervention set, feedback-geometry compatibility (manifest as the strict fake-mirror inversion at minimal scale; §1.5) separates into two operational perturbation families. At the **coordinate level** (C1), permuting the feedback vector destroys dimensional identity while preserving marginal statistics—and this is harmful at every tested scale, from 35 neurons to MNIST at 120k parameters. At the **geometric level** (C2), the clone's output preserves coordinate structure but carries a misaligned continuous geometry; the penalty here is scale-dependent. It is actively disruptive at small scales where independently trained models develop divergent representations, transitioning to a net-positive but still suboptimal signal under variable noise at the larger tested widths and on MNIST, while under static input it attenuates toward near-neutral (the mean stays slightly negative, but the per-width C2$-$A CI overlaps zero at $w{\geq}45$; App. H). This pattern is consistent with representation convergence at larger scales (Huh et al., 2024). The C1/C2 dissociation across

scale is itself informative: the coordinate-level component is the more robust phenomenon (its per-width CIs exclude zero at every scale), while the geometric-level component is regime-specific. Even at the scales we tested where C2 is no longer absolutely harmful, self-feedback retains a significant advantage (§ 3.6, § 3.9).

We register an explicit prediction: the BL−C2 gap may continue to narrow as scale increases and representational convergence (Huh et al., 2024) progresses, potentially reaching parity at scales we have not tested. Such a result would recast feedback-geometry compatibility as a scale-bounded phenomenon, without affecting the C1 coordinate-level penalty (which replicates at every scale tested) or the closed-loop residual from the static-input contrast. A related problem arises in multi-agent self-play: independently trained agents develop incompatible conventions that break during cross-play (Hu et al., 2020). Our setting is the single-agent, closed-loop version of this—the recurrent weights are tuned to their own model's output conventions during training.

The alignment decomposition (§ 3.5) shows why this sensitivity persists within the static open-loop family we tested. Progressively stronger static alignment models (30 to 17,925 parameters) recover most of the C2 degradation but saturate at $\sim 85\%$ recovery under variable noise, with a $3.7\times$ capacity increase from MLP-medium to MLP-large yielding no additional improvement. The residual therefore reflects a limit of static open-loop alignment for a closed-loop feedback task, not insufficient capacity. The alignment is calibrated in open-loop but evaluated in closed-loop, and static open-loop similarity does not guarantee dynamic closed-loop compatibility during iterative refinement; the saturation pattern follows from this distribution shift. Appendix G tests whether this is a representational limit or a covariate-shift effect directly: a supervised closed-loop adapter performs above Baseline and uses the donor (label-leakage diagnostic not triggered), but does so in a heavily saturated feedback regime atypical of the receiver's self-feedback. The saturation ceiling therefore constrains label-free static open-loop alignment specifically; supervised closed-loop adaptation closes the performance gap by operating the receiver outside its native regime.

## 4.3 Broader Connections and Implications

Our time-weighted loss shares a conceptual motivation with Chain-of-Thought prompting (Wei et al., 2022) (as scoped in §1.5): both allocate intermediate steps before the final-answer evaluation, granting the model additional processing without scoring intermediate exploration, although the underlying mechanisms differ (CoT is prompt-induced behavior at inference; our setting is a training-time loss reweighting). Unlike PonderNet or ACT, our approach requires no explicit halting mechanism—only a loss reweighting. The Group A necessity finding (no recurrence yields zero terminal-step gain in all settings) parallels work on self-referential meta-learning as a route to reducing explicit meta-optimization (Kirsch & Schmidhuber, 2022)—a necessity strictly bounded to single-pass terminal-step evaluation, distinct from post-hoc temporal aggregation of stateless outputs (§ 3.6).

Extrapolation from toy models to billion-parameter systems requires caution, but the MNIST extension (§ 3.9) provides a partial bridge: self-correction and the C1 coordinate-disruption penalty replicate at 120k parameters on an established image-classification benchmark, while the scale-bounded C2 inversion (C2 worse than no feedback) does not. The C2 penalty is scale-dependent—actively harmful at small scales, suboptimal but not absolutely harmful at MNIST scale—suggesting that the geometric-level component of feedback-geometry compatibility is strongest when model representations are most individualized.

The C1 result—that misaligned feedback is worse than no feedback—is a cautionary observation for systems that inject external or transformed signals into a pathway trained on self-generated feedback: feedback that deviates from the model's own output geometry may actively degrade rather than merely fail to improve. Precedents in adjacent domains observe conceptually analogous constraints: in knowledge distillation, stronger teachers can paradoxically produce worse students when the capacity gap is large (Mirzadeh et al., 2020). Whether activation-space compatibility acts as a similar bottleneck when incorporating test-time external critics at scale remains an open empirical question beyond the scope of this evaluation.

### 4.4  Limitations

1. **Scale and mechanistic specificity.** Thirty-five neurons performing pattern classification cannot capture the complexity of language or reasoning. The mechanistic analyses (decision-space visualization, $W_{\text{rec}}$ dissection, cosine divergence) are specific to this architecture; whether the same geometric signatures appear in larger recurrent systems is untested. The MNIST extension (§ 3.9) provides a partial bridge to 120k parameters, but a substantial gap remains to the scales at which iterative refinement is practically deployed. Both the prototype task and the MNIST extension are temporal-denoising problems with independent per-step noise; this differs structurally from the semantic iterative reasoning deployed at scale, and we make no claim that the two share underlying geometry.

2. **Effect size in context.** Under variable noise, the $+0.189$ correction gain corresponds to a $\sim 26\%$ relative improvement over the $t=1$ accuracy. Under static input, the gain is smaller but the primary claim—that feedback utility depends on geometric source—rests on the relative ordering across conditions, not on absolute gain magnitude. As noted in § 3.2, the parameter-matched feedforward model D$'$ achieves higher single-pass accuracy than the recurrent Baseline's final $t=3$ accuracy, confirming that recurrent refinement complements rather than replaces feedforward capacity.

3. **Alignment recovery structure.** The primary alignment models tested in §3.5 are static, open-loop, label-free mappers: trained on independently collected (donor logit, target logit) pairs from open-loop self-feedback runs and applied without adaptation during the closed-loop evaluation. Appendix G reports a complementary task-supervised closed-loop BPTT alignment ablation; that arm performs above Baseline and the donor contributes beyond an $x_t$-only control, but the closure is reached in a saturation-heavy regime atypical of the receiver's native self-feedback. The static-MSE saturation finding therefore characterizes label-free open-loop alignment specifically; whether a label-free closed-loop alignment scheme can bridge the gap without driving the receiver into this atypical regime remains open.

4. **Wrong-trajectory noise-sequence confound.** The wrong-trajectory control (§2.5) replaces feedback with the same model's output from a different, class-matched trial. Because each trial in the variable-noise regime carries an independent per-timestep noise sequence, the substituted feedback necessarily comes from a different noise realization than the input the receiver is currently processing. A small portion of the positive correction gain preserved under wrong-trajectory feedback (static $+0.057$, VN $+0.108$) may also reflect a beneficial uncorrelated-evidence ensembling effect from this independent noise pairing, rather than purely model-identity preservation. The relative ordering—wrong-trajectory > no-feedback > clone—is robust to this concern, but the absolute magnitude under wrong-trajectory should not be over-interpreted as a clean isolate of state-conditional mismatch.

5. **Angular vs. source-identity in the clone dissociation.** §3.7's three-condition cosine-divergence ordering (wrong-trajectory < same-class proxy < clone, in the same order as harm magnitude) confounds smaller angular distance with limitation 4's ensembling bonus in the wrong-trajectory cohort; source-identity vs. angular-distance therefore remains unresolved. A parametric angular sweep at fixed L2 norm would be required to separate them.

6. **Hyperparameter coverage of the C2 inversion.** The robustness sweep (§3.8) validates Baseline self-correction emergence across $\tau$ and $w_1$ configurations (40/40 VN configurations positive); the C2 absolute-harm inversion (C2 below both Group A and zero) additionally holds beyond the primary ($\tau=2.0$, $w_1=0$) configuration (C2 $= -0.122$): a targeted static spot-check ($N=20$) confirms it at $\tau=1.5$ ($-0.124$), $\tau=3.0$ ($-0.093$), and $w_1=0.1$ ($-0.131$), each with 16/20 seeds below Group A. Post-training rollout up to $T=20$ (no retraining, §3.8) confirms that C2 and C1 compound rather than self-correct at higher horizons, but retraining at a different $T$ was not evaluated. The qualitative inversion is therefore established across several training configurations and one rollout regime; its precise magnitude as a function of training hyperparameters is not exhaustively mapped.

## 5   Conclusion

In a 35-neuron recurrent network, we have identified and characterized *feedback-geometry compatibility*—the empirical compatibility relation between a recurrent receiver and the geometry of its own self-generated feedback (defined operationally in §1.5). Four thematic findings synthesize the evidence:

1. **A recurrent-specific residual emerges under static input and at MNIST scale.** The recurrent Baseline obtains +0.036 gain under static input, where any deterministic stateless aggregator on identical inputs must yield exactly zero by construction; the +0.036 magnitude itself depends on the deferred-commitment loss regime $w_1 < 0.3$ (App. D). Under variable noise the loop−ensemble residual at $\sigma \leq 0.5$ is +0.040 (low-band, $p = 0.007$, reported descriptively; convergence-dependent, Appendix J) and is not significant at high noise; at MNIST scale (120k parameters) the loop−ensemble residual is +0.0064. Self-correction replicates across hidden widths 10–245 and on MNIST, emerges even under truly uniform time-weighting ($w_1 = w_2 = 1.0$; all 40 models positive), and is not attributable to capacity or depth—the parameter- and compute-matched feedforward controls (D′, D″; §3.2) yield no terminal-step gain under static input.

2. **Feedback utility depends on geometric source and scale.** Feedback interpolation reveals smooth, broadly monotonic degradation from self-feedback to foreign contamination (zero-crossings at $\alpha \approx 0.4$–$0.55$), and per-trial L2-norm matching does not rescue the static clone harm (§3.7)—magnitude alone is not the explanatory factor. A same-model wrong-trajectory control points to a feedback-*source* dependence: the model's own output from a different class-matched trial preserves positive gain (static +0.057, VN +0.108) whereas clone feedback degrades it. Because this control also carries the noise-pairing confound of §4.4 limitation 4, we treat it as suggestive of partial model-identity vs state-conditional separation rather than a clean isolate. The coordinate-disruption (C1) penalty replicates at every tested scale; the strict *fake-mirror inversion* (clone feedback worse than the no-feedback baseline; C2) does not persist uniformly beyond minimal scale: under variable noise C2 transitions to a net-positive but still suboptimal signal at the larger tested widths and on MNIST, while under static input it attenuates toward near-neutral (mean slightly negative, but the per-width CI overlaps zero at $w \geq 45$). Self-feedback's *relative* advantage over clone feedback nonetheless holds at every tested scale—the per-width Baseline−C2 paired difference excludes zero at all widths and is +0.009 on MNIST ($p < 10^{-4}$). This C1/C2 dissociation across scale is itself informative—the coordinate-level component is the more robust phenomenon (its per-width CIs exclude zero at every scale).

3. **Static open-loop alignment saturates below full recovery under variable noise; closed-loop BPTT closes the gap in a saturated regime.** Progressively stronger static, open-loop, label-free alignment models (30 to 17,925 parameters) recover essentially all of the static-input clone-feedback degradation but saturate near $\sim 85\%$ under variable noise; a $3.7\times$ capacity increase from MLP-medium to MLP-large yields no further variable-noise improvement. A complementary closed-loop BPTT ablation (Appendix G) performs above Baseline—donor marginal +0.03 to +0.055 beyond an $x_t$-only control, with the $\tau$-scaled feedback path driven into a heavily saturated regime. The saturation ceiling thus constrains label-free open-loop alignment specifically; closed-loop task supervision can close the gap, but through this outside-regime route.

4. **Local conditioning does not independently dissociate the feedback-source conditions.** A relative condition number analysis at the feedback receiver fails to separate self, clone, wrong-trajectory, and norm-matched clone feedback pairwise. We report these overlapping distributions as a local diagnostic only; the local condition number does not account for the behavioral differences, so our mechanistic claims rely on the trajectory-level analyses in §3.5: in output-logit decision space, self-feedback moves trial mass across the decision boundary in the corrective direction (+5.2 pp) whereas clone feedback shifts it the opposite way (−9.1 pp) and increases over-correction.

All foreign-feedback comparisons here concern zero-shot substitution into a frozen receiver trained under self-feedback, on prototype and MNIST temporal-denoising tasks; we do not test joint co-adaptation to

donor feedback or language-reasoning systems. Within that scope, the central finding is that feedback utility depends on source and geometry, not on generic temporal integration alone: self-feedback produces the largest correction gain, coordinate disruption is broadly harmful across scale, and clone feedback is geometry- and scale-dependent. We offer this as a fully transparent minimal case study of feedback-geometry compatibility, with MNIST-120k and width-sweep evidence bounding where the absolute inversion does and does not hold.

## Reproducibility Statement

All code, data, and analysis scripts are publicly available. The primary 35-neuron experiments are implemented in pure NumPy and can be reproduced on a standard multi-core desktop computer (`https://github.com/softkorea/fake-mirror-effect`); an independent Py-Torch reimplementation of the 35-neuron architecture (§3.6, Appendix A.6; `https://github.com/softkorea/pytorch-35neuron-validation`) and the MNIST extension (`https://github.com/softkorea/mnist-feedback-contract`) are in separate companion repositories. A master pipeline script (`run_all.py`) executes all 35-neuron experimental phases in sequence. Raw experimental data for the 35-neuron experiments are included as CSV files; MNIST results are provided in the companion repository.

## Acknowledgments

The author thanks the anonymous reviewers and the Action Editor for their feedback.

## Conflict of Interest

The author declares no conflict of interest. This research was conducted independently without institutional or corporate funding.

## Ethics Statement

This work studies fundamental mechanisms of iterative refinement in a minimal artificial neural network. The system is a 35-neuron toy model performing synthetic pattern classification; it has no direct real-world application and poses no foreseeable societal risk. The research may contribute to longer-term understanding of self-referential processing in AI systems, which is relevant to AI safety and alignment research. No human subjects were involved. All experiments are reproducible from the code and data described in the Reproducibility Statement, which are publicly available.

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

# A    Implementation and Verification

## A.1    Pure NumPy Implementation

The primary 35-neuron implementation is in pure NumPy without deep learning frameworks, ensuring that every computation in the main analyses is directly inspectable and that no hidden optimizer behavior, automatic differentiation artifact, or framework-level abstraction influences those results. The PyTorch cross-validation (§A.6) is a separate companion implementation used only as a robustness check, and the MNIST extension (§3.9) is similarly implemented in PyTorch in a separate companion repository.

## A.2 Gradient Verification

Backpropagation through time was implemented as a 3-step unroll, with numerical gradient checking (relative error threshold $10^{-4}$) to verify correctness. Numerical gradient checking on a representative model (seed=0) yielded maximum relative errors of $6.57 \times 10^{-8}$ (static) and $1.48 \times 10^{-7}$ (variable noise) across all weight matrices, well below the $10^{-4}$ acceptance threshold.

## A.3 Overfitting Assessment

At the 1000-epoch budget the mean train–test accuracy gap at $t=3$ is $+0.077$ (static) and $+0.041$ (variable noise); held-out accuracy is rising in both settings and does not degrade out to five times the budget. The full convergence and overfitting analysis across training budgets (500–5000 epochs) is reported in Appendix I.

## A.4 Evaluation Noise and Reproducibility

Under variable noise, each evaluation generates independently sampled noise sequences per timestep. Consequently, per-seed correction gains may vary across runs even for identical trained models, while aggregate statistics (means, CIs) converge. All reported VN results use a single consistent evaluation run; re-running evaluation with a different random state will produce different per-seed values but stable, convergent aggregate summaries.

## A.5 Mechanistic Transparency

Three classes of analysis were employed: (1) direct $W_{\text{rec}}$ visualization and arithmetic decomposition, (2) decision-space visualization in the $(y_{\text{true}}, \max_{c \neq \text{true}} y_c)$ plane (Figure 4), (3) exact causal interventions (per-neuron knockout, wrong-trajectory substitution, cosine divergence null baselines across 4,000 trials).

## A.6 PyTorch Cross-Validation

An independent PyTorch reimplementation of the 35-neuron architecture was used to verify that the core phenomena are not artifacts of the NumPy implementation. Under He normal initialization (matching the NumPy codebase), all qualitative patterns replicate ($N=20$): Baseline gain is positive (static: $+0.059$, VN: $+0.156$), Group A yields zero gain, and both C1 and C2 produce negative gains (the variable-noise C2 cell is near-null at $-0.005$). Effect sizes differ from the NumPy results because PyTorch's random number generator produces different weight initializations for the same seed; the qualitative ordering is preserved. Under PyTorch's default Kaiming uniform initialization, which produces $\sim 6\times$ smaller initial weight variance, self-correction fails to emerge entirely—all conditions yield gain $\approx 0$. This initialization dependence is consistent with the finding in § 3.8 that emergence depends on training conditions: smaller initial weights reduce the feedback signal below the threshold needed for the recurrent pathway to develop during training (Table 3).

**Integration-control replication.** A separate PyTorch integration-control run tested the same no-feedback within-network ensemble across the noise sweep under He normal initialization, evaluated in double precision (float64) to match the NumPy reference. The low-noise loop$-$ensemble residual is positive (descriptive $\sigma \leq 0.5$ low-band residual $+0.045$, exploratory unadjusted $p = 0.002$; at the $\sigma=0.5$ operating point the recurrent Baseline gain is $+0.184$, E1 log-product $+0.133$, residual $+0.051$, $p = 0.003$) and is not significant at high noise, reproducing the (descriptive, convergence-dependent) regime structure of §3.6 in an independent implementation. This residual is precision-sensitive: under single-precision arithmetic the $\sigma=0.5$ residual attenuates $7.8\times$ to $+0.0065$, consistent with its descriptive, non-load-bearing status.

# B Alternative Trajectory Visualizations

The main-text Figure 4 visualizes the BL/C2 contrast in output-logit decision space, the two-coordinate decision view in which the binary correct-vs-distractor decision is sufficient and the decision boundary $y_{\text{true}} = \max_{c \neq \text{true}} y_c$ is exactly representable. For transparency we preserve the Hidden-Layer-1 PCA projection that appeared in earlier drafts of this work (Figure 8) as a *superseded diagnostic*. This view is a 2D

Table 3: PyTorch cross-validation ($N$=20). Correction gain under two initialization schemes. Source: companion PyTorch validation repository (`pytorch_validation_he_normal.csv` and `pytorch_validation_kaiming.csv`).

| Init | Baseline | Group A | C1 | C2 |
|------|----------|---------|-----|-----|
| *Static* | | | | |
| He normal | $+0.059$ | $0.000$ | $-0.095$ | $-0.048$ |
| Kaiming uniform | $+0.002$ | $0.000$ | $-0.001$ | $+0.001$ |
| *Variable Noise* | | | | |
| He normal | $+0.156$ | $+0.004$ | $-0.129$ | $-0.005$ |
| Kaiming uniform | $-0.000$ | $-0.001$ | $-0.003$ | $-0.003$ |

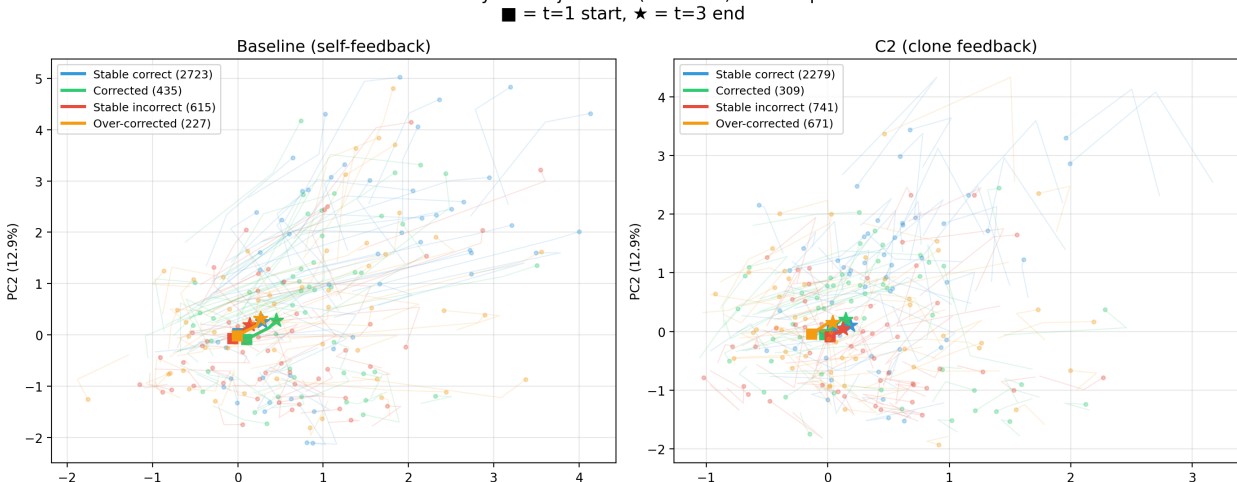

Figure 8: Original PCA projection of Hidden Layer 1 trajectories from earlier drafts of this work (seed = 0); each thin line is one test-trial trajectory $t=1 \rightarrow t=3$, with class-mean arrows overlaid. Preserved here as a transparency diagnostic only and *not* used as primary evidence in this paper; the decision-space view in main-text Figure 4 replaces it for the BL/C2 contrast, and the underlying mechanistic claims are documented numerically in §3.5 (cosine divergence, alignment decomposition) rather than from this projection.

projection of a 10-dimensional hidden state and compresses many orthogonal-to-class directions, producing dense overlapping trajectories that are difficult to read at a glance. The mechanistic claims of §3.5 are documented numerically (cosine divergence, alignment decomposition, Jacobian-conditioning) and visually in the decision-space view; we do not rely on Figure 8 as primary evidence.

## C    Control Architectures and Alignment Details

### C.1    Group D′: Parameter-Matched Feedforward

A feedforward network with an additional skip connection from input to output (50 weights), matching the total parameter count of the recurrent model (325 parameters).

## C.2 Group D″: Compute-Matched Feedforward

A deep feedforward network with 6 hidden layers of 10 neurons each (Input(10) → H1(10) → ... → H6(10) → Output(5)), matching the computational depth of the recurrent model's 3-timestep unroll. 715 parameters, approximately 650 multiply-add operations (vs. Baseline ~900 MACs across 3 timesteps).

## C.3 Affine Alignment

Both models were run with self-feedback on calibration data (400 samples × 3 timesteps = 1,200 logit pairs per pair); a linear map $\hat{y}_{\text{target}} \approx W \cdot y_{\text{donor}} + b$ (30 parameters: $5 \times 5$ weight matrix + 5 bias) was fitted to these pairs with the same regularized gradient-descent protocol as the stronger alignment family (Table 4) and applied during evaluation.

## C.4 MLP Alignment

A small MLP ($5 \to 16 \to 5$ with ReLU; 181 parameters) from donor logits to target logits, trained with MSE loss and gradient-clipped SGD (500 epochs, lr=0.01, max gradient norm 1.0), with a 20% sample-level validation split and best-epoch selection (patience 100).

## C.5 Stronger Alignment Family

Table 4 lists the alignment model configurations.

Table 4: Stronger alignment model configurations.

| Alignment Model | Parameters | Epochs | Learning Rate |
|---|---|---|---|
| Affine | 30 | 500 | 0.01 |
| MLP-small ($5 \to 16 \to 5$) | 181 | 500 | 0.01 |
| MLP-medium ($5 \to 64 \to 64 \to 5$) | 4,869 | 2,000 | 0.005 |
| MLP-large ($5 \to 128 \to 128 \to 5$) | 17,925 | 2,000 | 0.003 |

Calibration set: 400 samples (1,200 logit pairs). $R^2$ alignment quality of the gradient-fitted affine on the calibration pairs: $0.519 \pm 0.094$ (static), $0.530 \pm 0.107$ (VN). (The low $R^2$ measures donor→target logit-MSE fit, a different quantity from behavioral correction-gain recovery in Table 5: a coarse logit mapping can still recover most of the behavioral gain under static input.)

Table 5: Static-input alignment recovery ($N$=20), the complement to the variable-noise Table 2. Under static input a learned MLP aligner recovers essentially all of the clone-feedback (C2) drop (93–99%) while the affine aligner recovers most (76%); contrast the ~85% variable-noise ceiling. Recovery is the per-seed mean of $(\text{gain} - \text{gain}_{C2})/(\text{gain}_{BL} - \text{gain}_{C2})$; gains are mean [bootstrap 95% CI].

| Alignment | Static gain | Recovery |
|---|---|---|
| C2-raw (no alignment) | $-0.122\,[-0.175, -0.069]$ | 0% |
| Affine | $+0.011\,[-0.015, +0.038]$ | 76% |
| MLP-small | $+0.029\,[+0.004, +0.055]$ | 99% |
| MLP-medium | $+0.029\,[+0.002, +0.058]$ | 97% |
| MLP-large | $+0.029\,[+0.006, +0.054]$ | 93% |
| Baseline (self) | $+0.036\,[+0.014, +0.058]$ | 100% |

## C.6 MNIST Configuration

**Architecture.** Input(784) → H1(128) → H2(128) → Output(10) with output→hidden feedback (119,562 parameters), implemented in PyTorch. ReLU activations on hidden layers, linear output, and the same

temperature-scaled tanh feedback path $f_t = \tanh(y_{t-1}/\tau)$ as the 35-neuron model (Eq. 1–4, with the appropriate input/output dimensions).

**Dataset and per-timestep input.** Standard MNIST (60,000 training, 10,000 test images, $28\times28$ grayscale, normalized using the standard MNIST mean/std of 0.1307/0.3081). At each timestep $t \in \{1, 2, 3\}$ the model receives an independently noised view of the same image: $x_t = \text{image} + \varepsilon_t$, with $\varepsilon_t \sim \mathcal{N}(0, \sigma^2 I)$ sampled per pixel with $\sigma = 0.5$. This matches the variable-noise regime defined in §2.3; the static-input setting was not run at MNIST scale.

**Training.** 20 target models (seeds 0–19) and 20 donor models (seeds 100–119) trained for 30 epochs with mini-batch SGD (batch size 1024, lr = 0.01, no momentum) under the $\sum_i w_i$-normalized time-weighted cross-entropy loss $L = [\,0 \cdot L(t{=}1) + 0.2 \cdot L(t{=}2) + 1.0 \cdot L(t{=}3)\,]/1.2$ (normalized as in §2). Donor models are trained for the same 30-epoch budget under identical hyperparameters but on independent random initialization seeds.

**Evaluation.** The four feedback conditions (Baseline, Group A with $W_{\text{rec}}{=}0$, Group C1 with coordinate permutation of the feedback vector $f_t$, Group C2 with donor-output substitution into the feedback path) follow the 35-neuron protocol (§2.2), with one implementation difference: the MNIST C1 permutation is shared across the evaluation batch and redrawn per repetition and timestep (30 repetitions), whereas the 35-neuron implementation redraws it independently per trial and timestep. A per-trial variant evaluated on the same frozen checkpoints yields an indistinguishable result ($-0.021 \pm 0.007$, all 20 seeds negative, matching the batch-shared value at display precision; per-seed CSV in the companion repository), so the reported C1 effect does not depend on this choice. Paired target–donor evaluations use identical per-timestep noise sequences so output differences strictly reflect model identity.

**Determinism and reproducibility.** `torch.use_deterministic_algorithms(True)` and `cudnn.deterministic=True` were set; `CUBLAS_WORKSPACE_CONFIG=:4096:8` was exported before `import torch` as required for cuBLAS reproducibility on CUDA. Per-seed `torch.Generator` instances were used for both the DataLoader and the noise sampler, and SHA256 of every checkpoint together with the training-time environment (Torch / NumPy / CUDA / cuDNN versions) was recorded in a `manifest.json` accompanying the checkpoint set in the companion MNIST reproducibility repository.

# D   Statistical Details and Robustness

## D.1   Within-Model Pairing

Groups A, B1, C1, and C2 are post-hoc interventions on the same trained model (within-model pairing); Groups D, D′, and D″ are separately trained seed-matched controls.

## D.2   B2 Exclusion

B2 was excluded from the primary correction family a priori because its catastrophic feedforward degradation renders correction gain comparison uninformative.

## D.3   Emergence Heuristic

Mean gain $> 0$ with at least 60% of models showing positive gain. This criterion is deliberately liberal and intended only for exploratory grid characterization.

## D.4   Static Hyperparameter Grid

$w_1 \in \{0.0, 0.1, 0.2, 0.3\}$, $w_2 \in \{0.1, 0.2, 0.3, 0.5\}$, $\tau \in \{1.0, 1.5, 2.0, 3.0, 5.0\}$, 10 models each (800 total). Reported configuration ranked 9th/80 by mean gain.

Static gain depends substantially on $w_1$: mean gain = $+0.041$ at $w_1{=}0.0$ (78% positive), $+0.017$ at $w_1{=}0.1$, $+0.006$ at $w_1{=}0.2$, and $-0.001$ at $w_1{=}0.3$, indicating that static self-correction partly reflects deferred

computation from reduced early-timestep loss. In contrast, VN gain remains robust across all $w_1$ values (+0.100 to +0.193, 100% positive at every setting including $w_1=w_2=1.0$).

### D.5   Variable-Noise Grids

Targeted: $w_1 \in \{0.0, 0.1, 0.2, 0.3\}$, $\tau \in \{1.0, 1.5, 2.0, 3.0\}$, $w_2 = 0.2$ fixed (160 runs). Extended: $w_1 \in \{0.5, 0.7, 1.0\}$, $\tau \in \{1.0, 1.5, 2.0, 3.0\}$, $w_2 \in \{0.2, 1.0\}$ (240 runs).

### D.6   Static Noise Sweep

Under static input, correction gain was maximal at low noise (noise=0.1: gain $\approx +0.097$; noise=0.2: gain $\approx +0.072$) and declined monotonically at higher noise levels (noise=0.5: +0.036; noise=1.0: +0.008).

### D.7   Pseudoreplication Control

Each model used per-model data seeds (seed = model seed for training, seed + 500 for testing).

## E   Interpolation Data

Table 6 reports the full interpolation data. These models are trained at the primary protocol but evaluated on an independent per-model test draw (seed 1000+$s$, distinct from the primary's seed+500); the $\alpha=1.0$ self-feedback endpoint (+0.052) therefore differs modestly from the primary static Baseline gain (+0.036, §3.1) through test-set sampling alone. The large variable-noise gains are stable across draws while the small static gain is draw-sensitive, and the load-bearing quantity here is the within-experiment $\alpha$-trend, which is unaffected.

Table 6: Feedback interpolation results ($N$=20 models). $\alpha$=1.0 is pure self-feedback; $\alpha$=0.0 is pure contaminated feedback.

| $\alpha$ | Zero | Shuffle | Clone |
|---|---|---|---|
| 0.0 | +0.000 | −0.095 | −0.090 |
| 0.1 | +0.020 | −0.076 | −0.072 |
| 0.2 | +0.036 | −0.050 | −0.055 |
| 0.3 | +0.046 | −0.023 | −0.041 |
| 0.4 | +0.048 | +0.002 | −0.025 |
| 0.5 | +0.049 | +0.015 | −0.009 |
| 0.6 | +0.056 | +0.031 | +0.009 |
| 0.7 | +0.055 | +0.040 | +0.024 |
| 0.8 | +0.053 | +0.048 | +0.036 |
| 0.9 | +0.053 | +0.053 | +0.046 |
| 1.0 | +0.052 | +0.052 | +0.052 |

## F   Neuron Knockout Details

### F.1   Decoupled Knockout Methodology

Full neuron knockout simultaneously eliminates both the neuron's feedforward contribution to classification and its recurrent contribution to self-correction, confounding intelligence and correction effects. To decouple these, each Hidden Layer 1 neuron receives two separate single-pathway ablations, each compared against the intact model: zeroing only its feedforward inputs (its $W_{ih_1}$ column and bias) yields *intelligence importance* ($\Delta\mathrm{acc}_{t_1}$), and zeroing only its recurrent input (its $W_{\mathrm{rec}}$ column) yields *correction importance* ($\Delta\mathrm{gain}$). This avoids the baseline-collapse confound of full knockouts (§2.4).

## F.2 Hidden Layer 2 Caveat

For Hidden Layer 2 neurons (which have no direct recurrent input), full knockout was used; the intelligence-correction confound should be noted.

## F.3 Trial Classification Breakdown

Across all trials, the following classification breakdown was observed: corrected 10.9%, stable correct 68.1%, stable incorrect 15.4%, over-corrected 5.7%.

## F.4 Single-Neuron Knockout Heatmaps and Layer 2 Analysis

Figure 9 reports the seed-0 single-neuron knockout summary for all 20 hidden units. For Layer 1 (Panel A), intelligence ($\Delta\mathrm{acc}_{t_1}$) and correction ($\Delta\mathrm{gain}$) importance are decoupled by ablating only the feedforward-input column of $W_{ih_1}$ versus only the recurrent-input column of $W_{\mathrm{rec}}$, respectively. The H1 partition into intelligence-specialized (lower-right quadrant), correction-specialized (upper-left), and shared (upper-right) groups discussed in §3.5 is visible directly; the anomalous non-monotonic case h1_9 sits in the lower half (negative correction importance), visualizing the body finding that its recurrent input is non-constructive.

For Layer 2 (Panel B), the lack of direct $W_{\mathrm{rec}}$ input means we cannot cleanly decouple intelligence from correction: full knockout simultaneously removes the unit's feedforward contribution and any indirect downstream recurrent influence. The Layer 2 view is therefore exploratory and reported only as a full-knockout sensitivity check. With that caveat, we observe that the majority of Layer 2 units are intelligence-dominant ($\Delta\mathrm{acc}_{t_1} \gtrsim \Delta\mathrm{gain}$). Non-monotonic ablation cases parallel to h1_9 also appear in Layer 2: h2_6 shows negative intelligence importance (ablation slightly improves $t_1$ accuracy), and h2_7 and h2_9 sit in the upper-right region with both metrics positive. Under full knockout these patterns cannot be attributed differentially to recurrent versus feedforward contributions, and we therefore do not draw mechanistic conclusions from Layer 2 beyond the consistency of the non-monotonic phenomenon across layers.

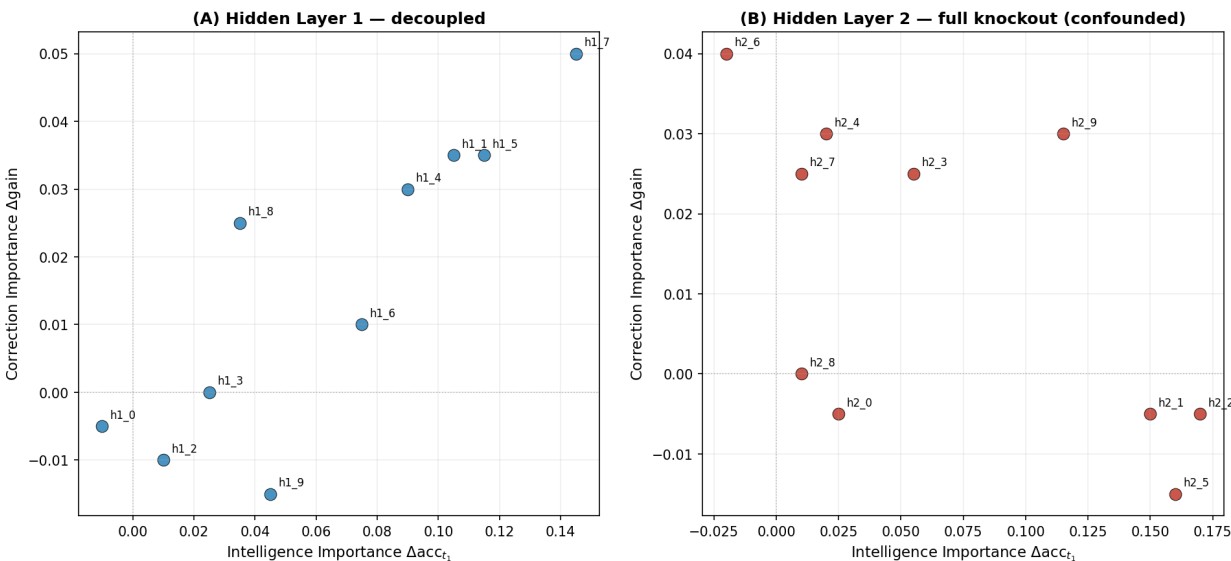

Figure 9: Seed-0 single-neuron knockout summary. **(A)** Hidden Layer 1, decoupled ablation: feedforward-input ablation ($W_{ih_1}$ column + bias) estimates intelligence importance ($\Delta\mathrm{acc}_{t_1}$); recurrent-input column ablation estimates correction importance ($\Delta\mathrm{gain}$). **(B)** Hidden Layer 2, full knockout: H2 has no direct recurrent input, so intelligence and correction effects are confounded and this panel is reported only as a sensitivity check. Across both layers, ablating individual units such as h1_9 and h2_6 yields negative effects on at least one metric, paralleling the non-monotonic finding in §3.5.

# G  Task-Supervised Upper Bound: Closed-Loop BPTT Alignment

## G.1  Setup and Protocol

The §3.5 alignment decomposition reports a *static, open-loop, label-free* family of MLP aligners trained to minimize the MSE between donor logits and target self-feedback logits on independently collected open-loop trajectories. Recovery saturates at ~85% within this family. As an open question raised by §4.4 limitation item 3 and the original §1.6 contribution 4, we now report an *in-loop, task-supervised* alignment ablation: an aligner trained end-to-end through the frozen receiver's recurrent unroll under task cross-entropy loss. Concretely, the aligner $A : \mathbb{R}^5 \to \mathbb{R}^5$ produces pre-tanh aligned logits and is trained via 3-step BPTT through the (frozen) target's forward pass with $w_1{=}0$, $w_2{=}0.2$, $w_3{=}1.0$ time-weighted cross-entropy loss (normalized by $\sum_i w_i$, as in §2) on the target's terminal output:

$$f_t^{\text{closed-loop}} = \begin{cases} \tanh(A(y_{t-1}^{\text{donor}})/\tau), & t > 1 \\ \vec{0}, & t = 1. \end{cases} \tag{5}$$

**Protocol.** Four aligner sizes match the static family parameter counts (affine 30; MLP-small 181; MLP-medium 4,869; MLP-large 17,925). PyTorch float64 throughout; Adam optimizer with learning rate locked via 5-seed pilot ($10^{-2}$ for all four sizes); calibration cohort 1,000 trajectories (800 train + 200 val) drawn fresh from the prototype distribution at VN $\sigma{=}0.5$; evaluation on the paper's frozen 200-sample test cohort with matched per-trial noise seeds. Early stopping on val loss with patience 100 (full 2,000-epoch budget). For paired H1 comparison with §3.5, we re-train static aligners *in the same PyTorch harness* on the same calibration cohort using the closed-loop arm's locked LRs (this is a matched-harness comparator, not an exact reproduction of the NumPy-primary static aligner in §3.5; median static-PyTorch recovery is ~84–92% across sizes, comparable to the §3.5 NumPy result of ~85% at MLP-large). The pairing for H1 is at the frozen-target-seed level, so the parity precondition (next paragraph) governs the structural identity of the closed-loop unroll across both arms.

**Label-free versus task-supervised objectives.** The static aligners in §3.5 are label-free in the operational sense that their objective is exclusively MSE on donor-logit/target-logit pairs; any label information is mediated through the frozen target's learned logit geometry. By contrast, the closed-loop arm here is task-supervised: cross-entropy provides a direct class-label gradient through the frozen receiver's unroll, placing label information directly in the training objective. This distinction is central to interpreting the result below.

**Parity precondition.** Before any aligner training, we verify that the closed-loop unroll matches the paper's primary protocol exactly. With the identity aligner $A(y){=}y$, closed-loop output matches the native C2 (clone) forward pass to bit-exact precision (atol $10^{-9}$, 20/20 target seeds). With the zero aligner $A(y){=}0$, closed-loop output matches the native no-feedback (Group A) pass to bit-exact precision. This confirms structural identity of the $t{=}1$ mask, $\tau$-scaled tanh path, and parameter loading.

**Diagnostic batteries.** Per (seed, aligner size) we record: (a) an $x_t$-*only control aligner*, approximately parameter-matched, taking only the input $x_t$ (no donor logits), designed to control for label decoding via the task-loss gradient path; (a′) a *bias-only mechanism diagnostic*, a 5-parameter aligner with no input at all (a constant 5-dim learnable vector applied at every timestep), designed to test whether the closed-loop alignment reduces to a constant bias injection through $W_{\text{rec}}$; (b) *shuffled-donor evaluation*, feeding the trained aligner donor outputs from mismatched test trajectories; (c) *zero-donor evaluation*, feeding zeros as aligner input; (d) *cosine to target self-feedback*, the cosine similarity between aligner output and target's actual self-feedback signal on matched inputs; and (e) a *bang-bang $\tau$ monitor*, the max absolute value of pre-tanh aligned logits per timestep (values $\gg 8$ indicate the $\tau$-scaled tanh has been driven into hard saturation).

Because the MLP-large aligner has 17,925 parameters versus the 325-parameter receiver, this closed-loop configuration should be treated as a task-supervised upper bound, not as a parameter-matched comparison to BL.

## G.2 Results

**Result.** All four aligner sizes (donor-fed) achieve mean correction gain on the frozen test cohort that is *above* the Baseline self-feedback gain (BL $= +0.184 \pm 0.059$):

Table 7: Closed-loop BPTT alignment results vs static-PyTorch (Static-PT) baseline (mean $\pm$ s.d. over $N$=20 target seeds). Donor marginal is the per-seed mean of (donor-fed gain $- x_t$-only gain), and may differ from the rounded column difference by up to 0.001.

| Aligner size | Closed-loop gain | $x_t$-only gain | Static-PT gain | Donor marginal |
|---|---|---|---|---|
| affine | $+0.199 \pm 0.057$ | $+0.144 \pm 0.056$ | $+0.135 \pm 0.067$ | $+0.055$ |
| MLP-small | $+0.203 \pm 0.061$ | $+0.170 \pm 0.075$ | $+0.159 \pm 0.063$ | $+0.033$ |
| MLP-medium | $+0.209 \pm 0.064$ | $+0.172 \pm 0.072$ | $+0.160 \pm 0.065$ | $+0.036$ |
| MLP-large | $+0.208 \pm 0.062$ | $+0.170 \pm 0.075$ | $+0.160 \pm 0.066$ | $+0.038$ |

Closed-loop significantly outperforms the static-PyTorch aligner at MLP-medium and MLP-large (paired exact Wilcoxon raw $p = 1.9 \times 10^{-6}$; Holm-corrected within the pre-specified $m$=4 family—two hypotheses at two sizes: *H1* (closed-loop gain > static-PyTorch aligner gain) and *H2* (closed-loop gain not worse than Baseline self-feedback), each at MLP-medium and MLP-large: $p_{\text{Holm}} = 7.6 \times 10^{-6}$). The paired closed-loop$-$Baseline difference is positive at every size ($+0.016$, $+0.020$, $+0.025$, $+0.025$ for affine/small/medium/large; mean-difference bootstrap 95% CI $[+0.009, +0.042]$ at MLP-large). A pre-specified two-one-sided test (TOST) for equivalence to Baseline within a margin $\delta = 0.02$—now adequately powered ($\delta$-pilot power 0.88 at this margin, above the 0.70 floor)—does not declare equivalence at any size, consistent with the closed-loop arm matching or exceeding Baseline rather than being statistically equivalent to it. The directional conclusion that closed-loop *exceeds* BL rests on the paired bootstrap CI, which excludes zero at every size.

## G.3 Diagnostic Flags

**Bias-only mechanism diagnostic.** A 5-parameter constant-bias aligner (input-independent) trained under the identical protocol achieves mean correction gain of only $+0.042 \pm 0.052$ across 20 seeds—well below $x_t$-only ($+0.170$ at MLP-large) and below BL ($+0.184$). The closed-loop "alignment" therefore is *not* a pure constant-bias injection; the aligner uses its input to produce trial-specific feedback. (The bias-only diagnostic is parameter-time-shared—5 parameters used at every timestep. A per-timestep bias parameterization (10 parameters at $t$=2, 3 with no input) is a tighter null we have not run; a positive result on that null would re-attribute the $+0.170$ $x_t$-only gain to per-timestep constant bias rather than to $x_t$-conditional readout.)

**Diagnostic flag status.** One of three pre-specified diagnostic flags triggers at the primary inferential sizes:

- **Label-leakage flag**: *not* triggered. The $x_t$-only control aligner (no donor input) is below the donor-fed aligner at every size; the donor marginal contribution is $+0.055$ (affine), $+0.033$, $+0.036$, $+0.038$ (MLP), positive in 16–18 of 20 seeds (paired Wilcoxon $p < 0.01$ each, robust to the sign test). The donor input therefore contributes beyond what $x_t$ alone provides, so the closure is not reducible to task-supervised label decoding from $x_t$.

- **Bang-bang $\tau$ flag**: triggered at all four sizes. The fraction of (trial $\times$ output-dim) pre-tanh elements in deep saturation (|aligned logit| > 4.0, beyond which $\tanh(\cdot/\tau=2) > 0.96$) ranges from 30% (affine) to 46% (MLP-large), all above the pre-specified 25% threshold; the per-size mean of the per-seed maximum absolute aligned logit at $t$=3 reaches 21–38 (individual seeds up to $\sim$142). How this compares to the receiver's native self-feedback distribution, and what the shift means, is detailed after the flag list.

- **Donor-decoupled flag**: *not* triggered. Donor perturbation at evaluation drops gain to $-0.21$ to $-0.24$ (shuffled donor, worse than C2 raw) and $-0.06$ to $-0.08$ (zero donor). The trained aligner is therefore donor-dependent—it has co-adapted to the specific donor, and (with the label-leakage flag cleared) the donor input contributes beyond $x_t$ alone, consistent with genuine use of the donor representation rather than donor-independent label readout.

**The saturation regime in context.** The receiver-training distribution, measured on 20-seed Baseline self-feedback rollouts on the same test cohort, is qualitatively different from the aligner's: mean $|y_t|$ is 1.46 at $t{=}1$ and 2.47 at $t{=}2$ ($\mathcal{O}(1)$), mean per-seed maximum $|y_t|$ is 7.77 at $t{=}1$ and 11.73 at $t{=}2$ (absolute maximum across all seeds: 11.0 and 19.9), and only 4.9% at $t{=}1$ and 20.7% at $t{=}2$ of self-feedback pre-tanh elements have $|y_t| > 4.0$. The aligner has therefore shifted 30–46% of the $\tau$-tanh feedback path from soft modulation toward saturated phase forcing, at peak magnitudes roughly 2–5× the receiver's own self-feedback, while 54–70% remain in soft modulation at $t{=}3$—a qualitative regime shift (not a wholesale binary code; cf. Interpretation bullet 3) establishing this as a regime encountered substantially less frequently under the receiver's own training distribution (20.7% deep-saturation fraction at $t{=}2$ vs 30–46% at $t{=}3$ in the closed-loop arm). The bang-bang threshold ($|$aligned logit$| > 4.0$) is calibrated to the receiver-training operating point $\tau{=}2.0$; the saturation interpretation is therefore stated at this $\tau$, and whether the controller would still drive saturation under a $\tau$-rescaled threshold at other $\tau$ values is not characterized here.

## G.4 Interpretation

With maximal task supervision, the closed-loop BPTT adapter *functionally bridges* the foreign-feedback gap: at every aligner size its gain is above the Baseline self-feedback gain (+0.208 vs +0.184 at MLP-large), and the closure is not reducible to label decoding, donor decoupling, or constant-bias injection. Three diagnostics jointly establish that the donor is genuinely used:

- **The donor contributes beyond $x_t$.** The donor marginal (donor-fed minus $x_t$-only) is positive at every size (+0.033 to +0.055; 16–18 of 20 seeds; paired Wilcoxon $p < 0.01$ each, robust to the sign test), so the label-leakage flag does not trigger: donor logits add gain over what the receiver's own input alone provides.

- **The aligner reads its input.** The bias-only aligner (5 params, no input) reaches only +0.042, far below both $x_t$-only and BL, so the closure is trial-specific feedback rather than constant-bias injection.

- **The aligner is donor-dependent.** Donor perturbation collapses the gain (shuffled $-0.21$ to $-0.24$; zero $-0.06$ to $-0.08$), so the trained adapter relies on the specific donor representation.

The bang-bang diagnostic characterizes *how* the gap is bridged: the optimizer drives the $\tau$-tanh feedback path into a heavily saturated, high-magnitude regime (30–46% deep saturation; peak magnitudes 2–5× the receiver's own self-feedback) rather than into a smooth donor-to-native mapping within the receiver's own feedback distribution.

**Synthesis.** The closed-loop arm is a supervised upper-bound: it makes donor information useful, but outside the receiver's native feedback regime. The $x_t$-only control already attains much of the gain (App. G diagnostic above), so the adapter is partly driving the receiver through task gradients, not translating donor feedback semantics. The $\sim$85% ceiling reported in §3.5 is therefore a label-free, static, open-loop property; closed-loop task supervision can close the gap, but via this outside-regime mechanism.

These diagnostics evaluate the minimal 35-neuron, $\tau{=}2.0$, $\sigma{=}0.5$ setting; scaling this closed-loop baseline to the MNIST cohort is left for future work.

## G.5 Reproducibility

**Reproducibility.** This appendix's experiments are implemented in PyTorch. The experiment driver `experiments/run_closed_loop_alignment.py` and the analysis script `experiments/analyze_closed_loop_alignment.py` ship in the main code repository and additionally require the PyTorch companion repository's network module (Appendix A.6) cloned alongside; outputs are stored in `results/closed_loop_alignment_summary.json` and the corresponding per-seed CSV files. The protocol and diagnostic flags are specified in this appendix.

Calibration and test cohorts use disjoint RNG seeds but are drawn from the same VN $\sigma{=}0.5$ distribution; distribution-shifted calibration is left for future work.

## H   Scale Verification and MNIST Data

### H.1   Scale Verification

Table 8 reports correction gain across hidden widths. The scale experiment evaluates all widths on a single shared test set (seed 999), rather than the per-model held-out sets (seed seed+500) used in the primary experiment; this reduces across-seed variance but does not bias the within-experiment condition contrasts. Consequently the smallest-width static Baseline gain ($w$=10, +0.054) differs modestly from the primary static gain (+0.036, §3.1) through the different test draw, whereas the larger variable-noise gain ($w$=10, +0.191) matches the primary (+0.189); the draw affects only the small static absolute values, not the C1/C2 ordering or the scale trend. In particular, the variable-noise $w$=10 C2−A interval in Table 9 ([−0.089, +0.002]) marginally overlaps zero on this shared-draw cohort; this is not in tension with the primary-cohort result (Holm $p$=0.025, §3.3), which uses the per-model held-out draw and the exact Wilcoxon test rather than this cohort's bootstrap interval.

Table 8: Correction gain across hidden widths (mean, $N$=20 models). Baseline is self-feedback; A is recurrent ablation; C1 is shuffled feedback; C2 is clone feedback.

| Condition | Hidden Width | Baseline | Group A | C1 | C2 |
|---|---|---|---|---|---|
| Static | 10 | +0.054 | 0.000 | −0.093 | −0.079 |
| | 20 | +0.064 | 0.000 | −0.108 | −0.064 |
| | 45 | +0.050 | 0.000 | −0.143 | −0.024 |
| | 245 | +0.029 | 0.000 | −0.289 | −0.020 |
| VN | 10 | +0.191 | +0.004 | −0.110 | −0.041 |
| | 20 | +0.169 | −0.018 | −0.174 | −0.021 |
| | 45 | +0.145 | −0.014 | −0.209 | +0.039 |
| | 245 | +0.145 | +0.008 | −0.319 | +0.076 |

Table 9: Per-width paired differences vs. Group A (no feedback) with bootstrap 95% CIs ($N$=20 seeds). BL−A is positive at every width (self-correction); C1−A is negative at every width with seed sign-count $\geq$ 19/20 (the coordinate-disruption penalty is robust across scale); C2−A is negative at small widths and turns positive by $w$=45 under variable noise (the scale-dependent inversion). Self-feedback's relative advantage (Baseline−C2) excludes zero at every width: static +0.133/ + 0.128/ + 0.074/ + 0.048 and VN +0.231/ + 0.190/ + 0.106/ + 0.068 at $w$=10/20/45/245. This is the per-width inferential complement to the means in Table 8.

| Regime | $w$ | BL−A | C1−A | C2−A |
|---|---|---|---|---|
| Static | 10 | +0.054 [+0.038, +0.069] | −0.093 [−0.117, −0.069] | −0.079 [−0.119, −0.041] |
| | 20 | +0.064 [+0.043, +0.085] | −0.108 [−0.126, −0.090] | −0.064 [−0.093, −0.039] |
| | 45 | +0.050 [+0.032, +0.071] | −0.143 [−0.160, −0.123] | −0.024 [−0.055, +0.006] |
| | 245 | +0.029 [+0.011, +0.046] | −0.289 [−0.315, −0.259] | −0.020 [−0.053, +0.012] |
| VN | 10 | +0.187 [+0.170, +0.206] | −0.113 [−0.143, −0.082] | −0.044 [−0.089, +0.002] |
| | 20 | +0.187 [+0.168, +0.208] | −0.156 [−0.181, −0.131] | −0.003 [−0.046, +0.036] |
| | 45 | +0.160 [+0.146, +0.173] | −0.194 [−0.213, −0.175] | +0.054 [+0.020, +0.085] |
| | 245 | +0.137 [+0.125, +0.149] | −0.327 [−0.354, −0.298] | +0.068 [+0.038, +0.096] |

### H.2   MNIST Per-Seed Data

Full per-seed gains are available in the companion MNIST repository's `paper_metrics.csv` (separate repository per the Reproducibility Statement; the local repository's `results/integration_control_mnist_cv.csv` contains the per-seed Baseline/E1 ensemble run for the MNIST integration-control comparison in §3.6).

### H.3 MNIST Training Dynamics

Table 10 tracks MNIST correction gain across training epochs.

Table 10: MNIST correction gain across training epochs ($N$=20 models). Baseline is self-feedback; A is recurrent ablation; C1 is shuffled feedback; C2 is clone feedback.

| Epoch | Baseline | Group A | C1 | C2 |
|---|---|---|---|---|
| 0 | $-0.002$ | $-0.001$ | $-0.001$ | $-0.001$ |
| 1 | $+0.013$ | $-0.001$ | $-0.018$ | $-0.032$ |
| 2 | $+0.018$ | $-0.001$ | $-0.021$ | $-0.003$ |
| 3 | $+0.018$ | $-0.001$ | $-0.023$ | $+0.005$ |
| 5 | $+0.021$ | $+0.000$ | $-0.023$ | $+0.011$ |
| 7 | $+0.021$ | $+0.001$ | $-0.021$ | $+0.013$ |
| 10 | $+0.019$ | $-0.001$ | $-0.022$ | $+0.011$ |
| 15 | $+0.022$ | $+0.001$ | $-0.021$ | $+0.013$ |
| 20 | $+0.020$ | $+0.001$ | $-0.021$ | $+0.012$ |
| 25 | $+0.019$ | $+0.000$ | $-0.021$ | $+0.011$ |
| 30 | $+0.019$ | $-0.001$ | $-0.022$ | $+0.009$ |

## I Convergence and Overfitting Diagnostics

All reported models use a uniform 1000-epoch budget. This appendix reports the convergence and generalization diagnostics behind that choice: correction gain as a function of training budget (Table 11) and the train–test accuracy gap (Table 12), each over 20 seeds.

### I.1 Correction gain vs. training budget

Table 11: Recurrent Baseline correction gain ($\mathrm{acc}_{t3} - \mathrm{acc}_{t1}$) vs. training epochs (20 seeds; normalized loss). The gain reaches its peak near the 1000-epoch budget and then drifts gently; it does not collapse.

| Setting | 500 | 1000 | 1500 | 2000 | 3000 | 5000 |
|---|---|---|---|---|---|---|
| Variable noise | $+0.086$ | $+0.189$ | $+0.183$ | $+0.181$ | $+0.177$ | $+0.171$ |
| Static | $+0.032$ | $+0.036$ | $+0.049$ | $+0.048$ | $+0.044$ | $+0.038$ |

### I.2 Train–test generalization gap

Table 12: Recurrent Baseline terminal accuracy ($\mathrm{acc}_{t3}$) on the training cohort vs. a held-out test cohort (disjoint seeds), and their gap, vs. training epochs (20 seeds). Overfitting would show held-out accuracy falling while the gap grows.

| Setting | | 500 | 1000 | 2000 | 5000 |
|---|---|---|---|---|---|
| Variable noise | train $\mathrm{acc}_{t3}$ | 0.787 | 0.943 | 0.993 | 1.000 |
| | test $\mathrm{acc}_{t3}$ | 0.735 | 0.902 | 0.951 | 0.963 |
| | gap | $+0.052$ | $+0.041$ | $+0.042$ | $+0.037$ |
| Static | train $\mathrm{acc}_{t3}$ | 0.714 | 0.861 | 0.942 | 0.984 |
| | test $\mathrm{acc}_{t3}$ | 0.655 | 0.784 | 0.838 | 0.841 |
| | gap | $+0.059$ | $+0.077$ | $+0.104$ | $+0.142$ |

### I.3 Training-budget choice

The 1000-epoch budget sits at a joint operating point: the correction gain has converged (it is at or near its peak for both settings), and neither setting is in a harmful overfitting regime (held-out accuracy is rising for variable noise and still rising at 1000 for static). Held-out accuracy does not degrade out to five times the budget for either setting. The static train–test gap does widen beyond $\sim$2000 epochs (training accuracy approaches ceiling while held-out accuracy plateaus), which is why we report the converged-but-pre-onset 1000-epoch point rather than a larger budget. The decomposition of the submission-to-revision protocol change (loss normalization vs. convergence) is given in Appendix J.

## J Submission-to-Revision Protocol Change: Convergence vs. Normalization

The revision adopts a uniform 1000-epoch budget and a time-weighted loss normalized by $\sum_i w_i$ (§2, Appendix I); the submitted version used an unnormalized loss at a 500-epoch budget. Two factors therefore changed at once. This appendix decomposes the change into its two components—loss normalization and training-to-convergence—using seed-matched data, so that the direction attributable to each is explicit. We report the variable-noise Baseline correction gain, the static Baseline gain, and the §3.6 loop−ensemble residual at the $\sigma$=0.5 operating point (Table 13).

Table 13: Decomposition of the submitted→revision protocol change (seed-matched, 20 seeds; all values at $\sigma$=0.5). Columns: submitted (unnormalized loss, 500 epochs); normalized loss at the same 500-epoch budget; and the revision operating point (normalized loss, 1000 epochs).

| Metric ($\sigma$=0.5) | unnorm., 500 ep | norm., 500 ep | norm., 1000 ep |
|---|---|---|---|
| Variable-noise Baseline gain | +0.119 | +0.086 | +0.189 |
| Loop−ensemble residual (§3.6) | −0.016 | −0.041 | +0.041 |
| Static Baseline gain | +0.042 | +0.032 | +0.036 |

The two components act in opposite directions. *Loss normalization alone* (first → second column, holding the budget at 500 epochs) lowers the variable-noise gain (+0.119 → +0.086) and leaves the loop−ensemble residual negative (−0.016 → −0.041): normalizing the loss removes the implicit 1.2× inflation of the effective learning rate, so at a fixed budget the network is slightly less converged. *Training to convergence* (second → third column, holding the loss normalized) raises the variable-noise gain (+0.086 → +0.189) and inverts the loop−ensemble residual from negative to positive (−0.041 → +0.041).

The §3.6 low-noise residual—the qualitative change in the low-noise analysis—therefore tracks *convergence*, not the loss normalization: the loss change on its own reduced the headline gain and left the residual negative, and the residual becomes positive only once the recurrent weights are trained to convergence (Appendix I). We accordingly report the converged values throughout and flag the direction of this change rather than attributing it to the loss redefinition. The submitted-protocol column reflects the original-submission run under the unnormalized-loss training regime, which was retired at the protocol revision; these are archival values, recorded here for the convergence-versus-normalization comparison, and are not regenerable from the released (normalized-loss) code.

**Sweep-wide protocol sensitivity.** This protocol sensitivity is not confined to $\sigma$=0.5; it spans the noise sweep in both directions. Pooled over the low band ($\sigma{\leq}0.5$), the loop−ensemble residual moves from +0.000 ($p$=0.91) under the submitted protocol (unnormalized loss, 500 ep) to +0.040 ($p$=0.007) under the revised protocol (normalized loss, 1000 ep); pooled over the high band ($\sigma{\geq}0.7$), it moves from −0.040 ($p$=0.009, the no-feedback ensemble exceeding the loop) to −0.009 ($p$=0.29, no reliable difference). The $\sigma$-structure of this revision-added integration control is therefore itself protocol-sensitive—which is precisely why the recurrent-specific claim is anchored on the static-input gain (§3.6, exactly zero for any deterministic stateless aggregator by construction, and positive at every protocol endpoint in this appendix) and the MNIST-scale residual (§3.9), and why we report the $\sigma$-sweep integration-control residuals descriptively rather than as load-bearing evidence.

