# OpenReview forum: "The Fake Mirror Effect: Foreign Feedback Disrupts Self-Correction in Minimal Recurrent Networks"
_TMLR — Accepted by TMLR_

### Review · Reviewer_fAhq · 2026-04-11

**Summary Of Contributions:**

The manuscript studies a foundational question about feedback mechanism. The authors research if the iterative refinement in recurrent neural networks is dependent on the source of the feedback. The authors construct a simple recurrent MLP with 35 neurons, and conduct experiments about self feedback, shuffled feedback and feedback from another independently trained model. The results show that when self feedback is replaced by clone feedback, the performance is worse. In some cases, the performance with clone feedback even drops below no feedback baseline. The manuscript also proposes explanations about feedback contract specificity. The topic exhibits a degree of novelty. Whether the model relies on its own output rather than equivalent external output during the iterative refinement process is worth exploring. This issue is related to several research directions, such as RLHF and knowledge distillation. Therefore, this work has potential influence.

In terms of mechanism analysis, the paper proposes the explanation of feedback contract specificity. Through the analysis of hidden trajectories, cosine divergence, and alignment decomposition, the authors show that this mismatch involves deeper geometric structural differences. Additionally, alignment experiments show that even with the introduction of strong linear or nonlinear mappings, performance can only be partially restored. This implies the existence of structural differences in closed-loop systems that cannot be compensated for by open loop mapping.

**Audience:**

Yes

**Audience Explanation:**

Yes, most readers may be interested. The paper focuses on feedback mechanisms in recurrent networks. The fake mirror effect is discussed. Self-feedback offers the strongest correction while shuffled feedback harms performance. The questions raised in this paper and conclusions are related to several research directions, including iterative refinement, representation alignment, knowledge distillation, etc. And thus it will attract researchers interested in mechanistic interpretability about feedback models. But the impact on larger scale models and applications in real world scenarios remains unclear. And thus the scope of the readers might be limited.

**Broader Impact Concerns:**

This work primarily studies a fundamental mechanism in models. It does not involve real world data or sensitive application scenarios. It does not pose any obvious direct ethical risks.

**Claims And Evidence:**

Yes

**Claims Explanation:**

The main claims are supported by strong evidence. The authors construct a minimal recurrent model and design systematic comparison experiments including self feedback, shuffled feedback, clone feedback, etc, to verify the fake mirror effect. The results are consistent across different settings.

The experimental design is rigorous and exhibits good causal isolation. Control groups and ablation are well-established. Feedforward controls, fake mirror and alignment controls are designed to quantify how much of the degradation is representation mismatch.

Besides, the paper also includes mechanism analysis (hidden trajectory analysis, cosine divergence, and alignment decomposition, etc.) to provide further support for the proposed feedback contract specificity. The explanations go beyond empirical observations.

However, the evidence is mainly based on small-scale models. Although authors provide MNIST experiments as a supplement, evidence of applicability for more complex models, such as transformers or LLMs is missing. Therefore, the conclusions are convincing in the current setting, but their generalizability remains uncertain.

**Requested Changes:**

Experiments are mainly based on small-scale models and simplified tasks. Although MNIST is provided, readers may still be interested in the fake mirror effect in more complicated large scale models. Authors have pointed out limitations of their research in section 4.4. Building upon this, the authors can further strengthen the discussion of these limitations and provide additional validation for the applicability of the proposed findings.

In addition, since the alignment experiments are limited to open loop mappings, it would be beneficial to discuss whether alternative approaches could mitigate this issue, and what impact such methods might have. It remains unclear if more expressive alignment strategies can further reduce the performance gap. Therefore, the claim that the mismatch reflects structural differences in closed loop systems may be vague to some extent.

---

> ### Author Response · Authors · 2026-04-12
> **Initial author response to Reviewer fAhq**
>
> Thank you for the thoughtful and constructive review. We appreciate your positive assessment of both the evidential support for the claims and the relevance of the paper to the TMLR audience.
>
> We agree with the two main points you raise. First, we agree that the strongest claims should be scoped more carefully with respect to generalization beyond the tested recurrent settings. In the revision, we will strengthen the discussion of limitations and clarify the scale-dependent interpretation of the results.
>
> Second, we agree that our current alignment experiments evaluate only the tested family of static open-loop mappings. In the revision, we will make this boundary more explicit and soften the interpretation accordingly. Our current results do not rule out the possibility that more adaptive or in-loop alignment strategies could reduce the residual gap further, and we will expand this discussion in the revised manuscript.
>
> Thank you again for helping us sharpen both the scope and the interpretation of the paper.

---

> ### Author Response · Authors · 2026-05-28
> **Author response to Reviewer fAhq**
>
> Dear Reviewer fAhq, thank you for the thorough and constructive review and for your positive assessment of both criteria. Your two requested changes pointed to genuine boundaries of our analysis, and the closed-loop alignment experiment your Requested Change 2 prompted (new Appendix G) gave a clean result that lets us delimit precisely what the §3.5 open-loop saturation does and does not show. Because OpenReview's per-comment 5000-character limit is shorter than a full point-by-point response, the complete reply is included in the supplementary as `responses/response_fAhq.pdf`. Headline summary below.
>
> - **Scale generalization (Requested Change 1).** The abstract, §3.9, §4.2, limitations, and conclusion now distinguish the robust C1 coordinate-disruption result (harmful across all tested scales) from the narrower C2-vs-no-feedback effect, which is scale- and regime-dependent. Under variable noise C2 transitions from absolute harm at minimal scale to a net-positive but still suboptimal signal at the largest tested width and on MNIST; under static input C2 attenuates toward near-neutral at the largest widths.
>
> - **MNIST clean-rerun with provenance (§3.9, supplementary).** The MNIST extension was rerun from scratch under deterministic settings in a clean companion repository, with per-checkpoint SHA256 hashes and a pinned environment recorded for provenance. This deterministic rerun is the source of the revised §3.9 numbers (Baseline +0.019, C1 −0.021, C2 +0.010; 20/20 seed-level directions). The full bundle is in the supplementary at `mnist-feedback-contract-submission/reproducibility/`.
>
> - **Alternative (non-open-loop) alignment (Requested Change 2 → new Appendix G).** We ran the closed-loop BPTT alignment ablation your comment pointed toward. A learnable aligner trained via BPTT through the frozen receiver's recurrent unroll under task cross-entropy loss achieves correction gain above Baseline at every tested capacity, and the donor input is load-bearing (donor marginal +0.03 to +0.06 beyond an $x_t$-only control; label-leakage and donor-decoupling diagnostics cleared). The closure is reached, however, by driving the receiver into a heavily saturated feedback regime atypical of its native self-feedback. We therefore scope the ~85% static-alignment ceiling reported in §3.5 to label-free, static, open-loop alignment specifically — not an absolute information-theoretic limit. The full diagnostic battery (bias-only control, shuffled-donor, zero-donor, bang-bang τ monitor) is in Appendix G.
>
> - **Scope of the §3.5 saturation claim narrowed (§4.4 item 3).** The static-MSE saturation finding is now characterised as a property of label-free open-loop alignment specifically; whether a label-free closed-loop alignment scheme can bridge the gap without driving the receiver into the atypical regime remains open and is flagged in limitations.
>
> - **PyTorch validation companion synced to the revised protocol** (1000-epoch, weighted-average loss). Files match the protocol cited in Appendix A.6 of the revised paper. The bundle is in the supplementary at `pytorch-35neuron-validation-submission/`.
>
> The full reply in `responses/response_fAhq.pdf` provides verbatim quotes, the Appendix G diagnostic results in full, and explicit pointers (section/line) for every change.

---

### Review · Reviewer_2HZg · 2026-04-27

**Summary Of Contributions:**

The paper discusses topics on recurrence, closed-loop feedback, and self-correction within a specific small-scale architecture. Experiments center around a simple setup: a small recurrent MLP learns to use its own previous logits as a feedback signal; if you permute or replace that feedback, performance drops. As the reviewer reads the paper, the "fake mirror effect" is the observation that when two independent recurrent MLP models are trained on the same task, replacing one models own feedback with the other incurs a bigger drop than with no feedback at all.

These disruptions are most damaging when feedback coordinates (input logits) are permuted. While "clone" feedback is harmful at a minimal scale, this geometric sensitivity attenuates as models get larger, though simpler coordinate disruption remains severe. The authors further include a wrong-trajectory control in which a model's feedback is replaced with its own output from a different trial, which preserves positive gain and dissociates foreign-model identity from generic state perturbation. Using static alignment to map one model's outputs to another recovers ~85% of the lost performance, with capacity scaling saturating at this level.

However, this reviewer finds it unclear why these findings should be considered unexpected, or to what extent the “fake mirror” effect is itself especially noteworthy. While the authors do somewhat address this, the contributions remain fairly broad and general compared to the results. Overall, the framing and claims around the results come across as somewhat stronger than the evidence presently supports.

**Additional Comments:**

It is fully possible that this reviewer has misunderstood or misinterpreted parts of the work, and I welcome the authors to engage in a constructive discussion to clarify the contributions and claims in their paper.

**Audience:**

Yes

**Audience Explanation:**

As it stands, I believe the paper is formulated around an interesting topic, but the findings still read as somewhat fleeting. As noted in the summary, many of the findings in the paper come across as expected behaviour. The narrative frames these results as novel discoveries, but they often read as confirmations of known behaviour placed in a mechanistic interpretability setting.

The core issue revolves around the conceptual framing of the paper. The authors do note in §1.4 that degradation under foreign feedback is unsurprising, and identify several structural features they consider non-obvious:

- that foreign feedback actively harms rather than fails to help,
- that permuting the feedback coordinates and substituting geometrically misaligned inputs is harmful in different ways,
- that fitting a mapping from one model's outputs to another only partially closes the gap, even as the mapping is given more parameters,
- that clone feedback becomes less damaging at larger model sizes, even though shuffled feedback remains harmful at every scale tested.

These are reasonably interesting findings. However, standard theory on numerical stability and distribution shifts tells us that injecting geometrically unaligned vectors into a non-linear closed loop will necessarily produce degradation. The saturation during alignment is observed only within a somewhat narrow scope, primarily through a limited family of alignment adapters (static, open-loop, MSE fitting). It is not yet clear that any of this requires an explanation beyond what the current theory predicts.

As it stands, this reviewer believes the manuscript needs to clarify the framing of the current experimental setup and hypotheses, and make a convincing argument for this particular methodological framing. This reviewer is open to the fact that this interpretation needs correction, but also points to the fact that the manuscript needs to credibly argue the key motivation for this choice.

However, the topic itself is clearly something that TMLR's audience could be interested in, even if the framing comes across as a little unconventional. I therefore lean on the guidelines, which state that "a reviewer that is unsure as to whether a submission satisfies this criterion should assume that it does".

**Broader Impact Concerns:**

The paper studies recurrence and feedback in machine learning, and does not touch on anything requiring a broader impact statement.

**Claims And Evidence:**

No

**Claims Explanation:**

As stated in the summary, the claims come across as somewhat grander than the evidence. The paper is a little broad in how it uses terminology. "Self-reference", 'fake mirror", and analogies to Chain-of-thought and RLHF are somewhat intertwined, which makes it difficult for a reader to really distill a concrete view of the subject matter. "Self-reference" reads as functionally equivalent to closed-loop feedback. If it is, or even if it is something else, then readers would appreciate a definition.

In the variable-noise task, the gains look like relatively straightforward evidence accumulation over repeated noisy observations. The authors acknowledge this, but the narrative then slips back into self-generated reasoning feedback. The issue this reviewer has is that the lines are blurred, and the authors do not provide a clear nomenclature with definitions of what they mean when jumps are made between more canonical terminology and more evocative phrasing. The static-input result (+0.042, CI excluding zero) does provide some evidence for a non-integration component, since the same input is presented at every timestep and the model therefore cannot improve simply by combining independent noisy observations over time.

However, the magnitude is small relative to the variable-noise gain, and the paper's framing tends to emphasize the larger number. More broadly, these findings suggest that much of the reported effect may reflect numerical sensitivity to an induced distribution shift rather than a clearly novel phenomenon. While OOD language is also somewhat notorious for being "difficult to quantify", it is at least more standard practice and can help readers form stronger anchors when reading the paper.

Similarly, the result on "foreign feedback is worse than no feedback" comes across as less mysterious than the authors make it out to be; even simple models learn a geometry or manifold. If a layer expects a specific coordinate structure, then strongly perturbing that will necessarily be damaging. The wrong-trajectory control (§3.5) does push back against the simplest version of this critique, since same-model wrong-trial feedback preserves positive gain while clone feedback does not. However, the cosine divergence values reported in Figure 4 differ substantially between these conditions (0.386 for resampled self-feedback vs. 0.701 for clone), so the dissociation may reflect magnitude rather than a qualitative distinction between self and foreign sources. Robustness analysis is a well-established mathematical discipline, and the paper does not rely much on tools and established theory in the field. How sensitive are recurrent updates to feedback perturbations? Measuring something like a relative condition number will say a lot more than pointing to the fact that learned coordinates are not arbitrary, and that perturbations deteriorate outputs.

As this reviewer reads the results, the "self-correction" gain in the variable noise task (+0.119) is much higher than in the static task (+0.042). This suggests that a substantial portion of the effect reflects standard temporal integration of noisy observations. The "fake mirror" harm may therefore be a byproduct of adapting a 35 neuron model to its own specific geometry. Again, without a more rigorous mathematical definition of what separates the "contract" from standard joint optimization, the claims made in the paper become a little too speculative.

**Requested Changes:**

The following points would serve to strengthen the work:

1. The motivating hypothesis and framing should be clarified. The manuscript should go a little further in explaining what is expected to be surprising in the current setup. We know that foreign feedback degrades performance. The question is why the particular structural features identified in §1.4 should be interpreted as non-obvious.
2. The conceptual framing and nomenclature could be more precise. In particular, terms such as “self-reference,” “self-correction,” and “feedback contract” should be clearly defined and distinguished from more standard notions such as closed-loop feedback, evidence accumulation, and co-adaptation under joint optimization. Other terminology could be used here, but it should be grounded in clear definitions.
3. The perturbation analysis should be strengthened with a quantitative local analysis, alongside the existing intervention cases. The paper would benefit from standard robustness tools, such as sensitivity analysis, relative condition numbers, or even local linearization, to distinguish simple coordinate mismatch from effects that depend more specifically on the recurrent trajectory.
4. The distinction between evidence accumulation and "self-correction" could be clearer. Since the variable-noise gains are larger than the static gains, the work should make attempts isolate how much of the reported effect comes from temporal integration of noisy observations versus something more specific to the feedback mechanism. Having some evidence that removes this as a confounder would strengthen the claim.
5. The claims should be grounded more carefully in the evidence presented. For instance, the degradation induced by clone feedback appears to weaken or disappear across scales, while degradation from permutation appears more consistent. The paper’s conclusions should reflect that this part of the effect is not uniformly robust. More broadly, some claims come across as overly general relative to the results shown, as stated in the summary.

---

> ### Author Response · Authors · 2026-04-29
> **Initial author response to Reviewer 2HZg**
>
> We thank the reviewer for the careful and constructive review. We appreciate the clear statement of the main concern: that the present manuscript should distinguish more sharply between standard closed-loop feedback / distribution-shift explanations and the more specific "fake mirror" framing.
>
> We agree that the revision should make the terminology and scope more precise. In particular, we will define how we use "self-correction", "self-reference", and "feedback contract", and we will distinguish these from evidence accumulation, closed-loop feedback, and ordinary co-adaptation under joint optimization. We will also revise the framing so that analogies to chain-of-thought and RLHF are clearly scoped rather than load-bearing, and we will articulate more carefully which features of the present results require more explanation than a generic distribution-shift or co-adaptation account alone provides, rather than relying on analogy.
>
> We also agree that the evidence-accumulation issue in the variable-noise setting needs to be treated more explicitly. The static-input result was intended to isolate the non-integration component, but the revised manuscript should quantify how much of the variable-noise gain can be explained by temporal integration alone. We will evaluate a non-recurrent integration control and scope the claims accordingly.
>
> We further agree that the perturbation results should be connected more directly to standard robustness tools. We will examine a local sensitivity / linearization analysis around the feedback receiver, including relative condition numbers and magnitude-matched perturbations where appropriate, and we will report the results whether they support a trajectory-specific interpretation or require a more conservative framing.
>
> Lastly, we will temper the scale-related claims, especially the distinction between the consistently harmful C1 coordinate-disruption effect in our tests and the scale-dependent C2 clone-feedback effect. A full point-by-point response and revised manuscript will be uploaded after all reviews have been received.

---

> > ### Comment · Reviewer_2HZg · 2026-05-26
> >
> > Dear authors,
> >
> > From your comment, I understood your rebuttal as a general acknowledgement of the reviews, with an intent to revise the manuscript. While your rebuttal seems to indicate that you agree with the points made by reviewers, it seems no revision with changes has yet been submitted. As we are about to submit our final recommendations, I would encourage you to make the changes concrete by submitting a revised version of the manuscript.

---

> > > ### Author Response · Authors · 2026-05-27
> > > **Revision upload timeline**
> > >
> > > Dear Reviewer 2HZg,
> > >
> > > Thank you for the reminder. We agree. The revised manuscript and point-by-point response materials are in final preparation, and we will upload the complete revision package (revised PDF, responses, and cover note) by 29 May 2026.

---

> ### Author Response · Authors · 2026-05-28
> **Author response to Reviewer 2HZg**
>
> Dear Reviewer 2HZg, thank you for the careful and substantive review. Your central concern — that the framing should be brought closer to what the evidence directly supports — was well taken, and we have used the revision to scope each claim more precisely against the actual data. Because OpenReview's per-comment 5000-character limit is much shorter than a full point-by-point response, the complete reply is included in the supplementary as `responses/response_2HZg.pdf`. Headline summary below; full reply addresses each of your five requested changes plus the specific suggestions on temporal integration, magnitude, and alignment scope.
>
> - **Claim scope and load-bearing analogies (Requested Changes 1 and 5).** New §1.5 *Terminology and Scope* defines every paper-specific term operationally and includes a one-to-one structural-mapping table with an explicit "Limits of analogy" column scoping the chain-of-thought analogue to fixed-step categorical classification and the 35-neuron-to-MNIST band only. The claim hierarchy and multiplicity structure are stated explicitly with a *Descriptive analyses and anchoring* statement identifying which results are descriptive vs. confirmatory.
>
> - **No-feedback within-network ensemble control (Requested Change 4, §3.6).** A new integration control decomposes the variable-noise gain into a temporal-integration component (post-hoc E1 aggregation of the same network's per-step outputs) and a closed-loop residual. The recurrent-specific claim is anchored on two contrasts that do *not* depend on this convergence-sensitive comparison: the static-input contrast (any deterministic stateless aggregator yields zero gain by construction) and the MNIST-scale loop−ensemble residual.
>
> - **Local diagnostics (Requested Change 3, §3.7) + new measurements (§3.3).** Added Jacobian condition-number analysis, per-trial norm-matched clone feedback, and additional cosine-divergence measurements. A donor-capability control confirms the clone is capability-matched (donor vs. target t3 paired Wilcoxon p=0.53), and an agreement-stratified C2 analysis confirms that the C2 harm survives on the t1-agreement subset — so the penalty is neither a capability deficit nor reducible to class-level disagreement. As these local diagnostics do not pairwise dissociate the feedback conditions, the mechanistic interpretation rests on the trajectory-level analyses.
>
> - **Scale-claim scoping (Requested Change 5).** The robust C1 coordinate-disruption penalty (every tested scale) is now explicitly distinguished from the scale- and regime-dependent C2 absolute harm: under variable noise it transitions to a net-positive but still suboptimal signal at the largest tested width and on MNIST, while under static input it attenuates toward near-neutral at the largest widths. Reflected in abstract, §3.9, §4.2, limitations, and conclusion.
>
> - **Post-submission protocol standardization.** During revision we standardized the submitted unnormalized weighted-sum time-weighted loss to a weighted-average form and trained all primary 35-neuron configurations to 1000 epochs. Appendix J decomposes submitted-vs-revised changes including a low-band sign flip in the convergence-sensitive §3.6 noise sweep. Qualitative core conclusions (static-input recurrent gain, minimal-scale C1/C2 negativity, 40/40 uniform-weight emergence, MNIST-scale residual) are preserved.
>
> The full reply in `responses/response_2HZg.pdf` provides verbatim quotes from the revised paper, the additional discussion the magnitude/angular and saturation-scope concerns warranted, and explicit pointers (section/line) for every change.

---

### Review · Reviewer_HJE3 · 2026-05-06

**Summary Of Contributions:**

The paper studies how feedback in recurrent networks contributes to learning through a minimalist architecture and various interventions on a simple training pipeline. The authors find that corrupting the feedback in the recurrent network degrades the performance even if the feedback comes from a statistically equivalent source. They try to bridge that gap both with a simple interpolation with true feedback and by trying to map the true feedback with an auxiliary network. In all small scale cases, the experiments show what they call a "feedback contract" necessity. At larger scale, the results do not hold as well.

**Audience:**

Yes

**Audience Explanation:**

The approach of the paper is original to me and while the presentation is extremely obscure to me it may appeal to peers of the authors.

**Broader Impact Concerns:**

None foreseen

**Claims And Evidence:**

Yes

**Claims Explanation:**

The paper strives to give statistically significant results. There may be slight comments about e.g. the training procedure and most importantly the presentation but overall the results appear sufficiently supported.

**Requested Changes:**

The results seem interesting, but the paper is barely readable in my opinion. I may not be the target audience but enlarging the targeted public would ensure the proper dissemination of the results. Right now, to me, it does not make sense to publish the paper before some consequential work is put into its clarity for a larger public.

Below are my comments. My main issue is that the base experimental setup is not clear. We need an illustration and some proper mathematical definitions.
- The abstract is not an abstract, it is a conclusion. Abstracts should be readable by a large audience and invite the reader to dive into the results. The current abstract does not.
- Add illustrations for the main experimental setup: the recurrent network with its mlp architecture, the feedback mechanism, how is it corrupted etc...
- Consider maybe make a one-to-one map between some of the tangential fields and the experiments, for example chain of thoughts.
- In 1.2, you mention C1, C2 etc... while all this has not been presented. This reads less as an introduction than a discussion section.
- You may consider putting the prior work at the end since the reader will anyway skip it until he/she understood the experimental setup and the results
- Experimental setup clarifications:
  - Why do you use tanh on the output? It may generally cut gradient flow. How did you select the temperature to avoid that?
  - Why did you use pure numpy to implement this and not a known automatic differentiation framework?
  - Where does this task come from? It seems extremely specific. What are "prototypes"? Why is there inter-class ambiguity? What is it suppose to model? The authors mentioned links with chain of thoughts etc... maybe it would be great to use such connections to give more details on the experimental setup here.
  - What is actually the data? How can this not be detailed?
  - The authors should add more mathematical formulations. In the text, some loose notations (like $x_t, y_t, protototype_k$ are used: please give proper definitions. Ambiguity does not serve the presentation.
- 2.2.1.Group A: "All recurrent weights zeroed post-training": I don't understand aren't all weights recurrent if the network is a recurrent network? A picture, or proper mathematical definitions would have clarified this.
- 2.2.3 What is exactly the feedback vector. Once again a mathematical definition or an illustration would help. It seems to me that at the end, if we had the mathematical definitions, we could identify what objectives each setup is training. It would tremendously help understand the results.
- "Wilcoxon signed rank tests": explain completely the test, the procedure, and how to read the p values that are returned. Currently it is rather pointless to make all this rigorous effort only not to explain why.
- VN refers to Variable Noise I suppose but you did not explained it.
- "Because stateless models processing independent noisy inputs have an expected gain of zero by symmetry" -> Why? Do you have a reference?
- Avoid teasing results that appear later as done at the end of 3.1
- Section 3.5
  - Give formula for cosine divergence.
  - What is "same-class resampled feedback" ?
  - "At the seed level" -> what seed?
  - "dose-response relationship" -> what is that?
  - Figure 3 is simply looks like squiggles, it's impossible to parse.
  - "decoupled single neuron knockouts" -> what is that?
  - "Hidden Layer 1 partitioned ... groups" -> how do you know that?
- Detail setup with MNIST. That's actually the setup that could be the easiest for the reader to understand so full details could give an illustration to the whole experimental setup.
- Why did you choose SGD? RNNs are much better trained with Adam. Also 200 epochs seem to be a long time but as we don't know what is the data, it's hard to grasp.
- "Unlike PonderNet or ACT" -> give details of these competitors.

Questions:

---

> ### Author Response · Authors · 2026-05-06
> **Initial author response to Reviewer HJE3**
>
> Thank you for the careful and constructive review. We appreciate the positive assessment of the evidence supporting the claims, and we take seriously the central concern that the manuscript should be much easier to follow for a broader audience.
>
> We agree that the experimental setup needs to be presented more directly. In the revision we will add a schematic figure of the recurrent MLP architecture, the feedback receiver, and the main feedback perturbations; give the recurrent weight partition and feedback vector explicit mathematical definitions; and include a conceptual walkthrough of a representative trial of the synthetic prototype task, detailing what the task models, what the input and prototypes look like, how inter-class ambiguity is operationalized at the dataset level, and the mathematical form of the model's input at each timestep.
>
> We also agree that the abstract should invite a broader audience rather than read as a conclusion, that abbreviations such as VN should be defined at first use, and that several terms need clearer inline definitions or a glossary. We will add definitions for the cosine-divergence calculation, same-class resampled feedback, seed-level aggregation, decoupled single-neuron knockouts, dose-response language, and Hidden Layer 1 partitioning. We will also expand the brief comparison to PonderNet and ACT, explain our empirical rationale for the choice of SGD and the 200-epoch budget given the data size, and add a short methods paragraph providing standard methodological context for the Wilcoxon signed-rank procedure and the reported p-values.
>
> We will revise the ordering of the introductory material so that the experimental setup and synthetic data are introduced before condition labels (such as C1 and C2) and broader analogies or related-work connections. We will also soften forward-pointers to later results, replace the "expected gain of zero by symmetry" wording with the empirical integration-control analysis that quantifies the stateless baseline directly, improve the readability of the trajectory visualization, add full setup details for the MNIST experiment, and clarify the chain-of-thought analogy as high-level motivation while making explicit where the analogy does and does not map onto our experimental setup.
>
> A full point-by-point response and revised manuscript incorporating these clarity changes will be uploaded together.

---

> ### Author Response · Authors · 2026-05-28
> **Author response to Reviewer HJE3**
>
> Dear Reviewer HJE3, thank you for the review — the comments named the specific places where the paper's exposition was getting in the way of the science, and we agree on essentially every point you raised. Because OpenReview's per-comment 5000-character limit is shorter than a full point-by-point response, the complete reply (mapping every numbered comment to a specific revision change, Comment 1 through Comment 16) is included in the supplementary as `responses/response_HJE3.pdf`. Headline summary below.
>
> - **Redesigned decision-space figure (Figure 4; Comment 13).** The output-logit decision-space view replaces the dense Hidden-Layer-1 PCA panel as the primary figure. It uses the two-coordinate decision view in which the binary correct-vs-distractor decision is sufficient and the decision boundary $y_{\mathrm{true}} = \max_{c\neq\mathrm{true}} y_c$ is exactly representable, with three readable panels (corrected trajectory, distribution shift, trial-outcome taxonomy). The Hidden-Layer-1 PCA projection is preserved as a *superseded diagnostic* in Appendix B for transparency.
>
> - **Architecture-and-feedback schematic (new Figure 1; Comments 2 and 6).** A labelled diagram of the Input(10) → H1 → H2 → Output(5) network with the explicit output→hidden feedback path, distinguishing self-feedback (Baseline) from the four ablation conditions (A, C1, C2, wrong-trajectory). A *Setup at a glance* paragraph and a worked single-trial walkthrough (seed 0, true class 3) accompany §2.1.
>
> - **First-use definitions of previously undefined terms (Comments 10, 13).** Operational definitions added for: variable noise (§2.3), cosine divergence (with formula, §3.5), same-class resampled feedback (§2.5, §3.5), decoupled knockouts (§3.6, Appendix C), hidden-layer partitioning ($W_{ih1}$ vs. $W_{rec}$, §2.1, Eq. 1), seed-level aggregation (§2.6), $\tau$-scaled tanh feedback gate (§2.1).
>
> - **Explicit notation and conventions (Comments 6–8).** Notation table integrated into §2.1 and §2.6: $f_t$, $y_t$, $W_{rec}$, $W_{ih1}$, $\tau$, $T$, $\sigma$, gain, correction gain. All test-set seed conventions documented (primary: seed+500; wrong-trajectory: seed+1000; etc.).
>
> - **Plain-language explanation of the Wilcoxon procedure (Comment 9).** §2.6 now explains the exact signed-rank procedure step by step in plain English, the zero/tie tolerance convention (1e-9), and the multiplicity structure (Holm families pre-specified per claim, $m$ values explicit). Same for the bootstrap CI procedure.
>
> - **Other readability changes (Comments 1, 3–5, 11, 12, 14–16):** the abstract's structure rewritten so each sentence carries a specific operational claim; §1.1 motivation tightened; §3 result subsections reorganised so each headline result has its own subsection with a one-sentence summary, the numerical evidence, and a pointer to the relevant table/figure.
>
> The full reply in `responses/response_HJE3.pdf` walks Comment 1 through Comment 16 with verbatim quotes from the revised paper and section/line pointers.

---

### Author Response · Authors · 2026-07-03
**Camera-ready: summary of changes from the accepted version**

Dear Action Editor,

Alongside the camera-ready files, we would like to summarize transparently how the camera-ready differs from the accepted (May 28) manuscript. Beyond the standard de-anonymization, we used the camera-ready window to complete a systematic audit of the manuscript against the released code and data, including an independent full-pipeline reproduction on a separate machine. All resulting corrections are accuracy and consistency fixes: no primary experiment was added, none of the paper's claims or conclusions was strengthened, and several corrections make reported values or claim scopes more conservative. The primary results (Table 1), all statistical conclusions, and the abstract's conclusions are unchanged; the abstract's scale-scope wording was tightened in the conservative direction (item 4 below).

The material corrections are:

1. A small number of secondary-paragraph values that still reflected the pre-revision training protocol (the protocol change itself is documented in Appendix J) were re-synchronized to the released data: the static-input sweep summary (emergence 68% → 59% of grid configurations, now described as "more limited"; configuration rank 9th/80; per-weight means), the timestep-extension rollout accuracies (metric now defined in-sentence), the relative-improvement figure in the Limitations (~18% → ~26%), and the appendix alignment-fit $R^2$ (0.449/0.465 → 0.519/0.530).

2. The prototype table in the §2.1 setup box was corrected to reproduce exactly the prototypes generated by the released code (the typeset table showed a different leakage pattern; all computations always used the code, so no result is affected), and the worked-example logits in the same box were regenerated under the current protocol.

3. Protocol descriptions were aligned with the implementation where the two had drifted: the affine aligner is gradient-fitted on a 400-sample calibration set; the shuffled-feedback permutation is redrawn at each timestep (Figure 1's caption now matches §2.2 and the code); the MNIST shuffled-feedback implementation difference (batch-shared permutation) is disclosed together with a robustness check showing a per-trial variant yields an indistinguishable result; the Appendix G pilot learning rate and matched-harness recovery range were corrected to the released artifacts; training dynamics were tracked at 16 checkpoints.

4. A few edits tighten scope in the conservative direction: the feedforward-control claim is now scoped to static single-pass evaluation; the largest-width clone-feedback transition is stated as w ≥ 45 rather than only the largest width; two values were corrected against the released data (a single-neuron knockout importance and the recurrent-weight sign consistency, 0.559 → 0.563); a donor-control significance statement was weakened to the version robust across tests (p < 0.01, robust to a sign test); and a single-precision re-evaluation of the appendix cross-implementation check was added, showing the descriptive loop-ensemble residual attenuates under reduced precision, a conservative robustness note.

5. Minor wording, rounding-display, spelling (unified to American English), and cross-reference-robustness edits throughout, with the hedging and scoping language introduced during review preserved.

For verifiability, every reported number in the camera-ready is automatically re-derived from the released data by a 479-check verification gate included in the public code release, and the full pipeline was independently reproduced on a separate machine, with a single condition-number file differing only by $\le 10^{-7}$ floating-point round-off that affects no reported value. An itemized change list is available on request.

The camera-ready submission also includes the three public code repositories cited in the paper and a short supplementary video, per the acceptance email's encouragement.

Thank you for handling our submission.

Best regards,

Sungmoon Ong

---

> ### Comment · Action_Editor_7acL · 2026-07-07
>
> Thank you for the summary and updates!

---

### Decision · Action_Editor_7acL · 2026-06-18

**Recommendation:** Accept as is

**Additional Comments:**

The recommendation is based on the reviewers' comments, the action editor's evaluation, and the authors’ response.
This paper studies the effect of foreign deedback on self-correction for recurrent neural networks. All reviewers find the studied setting novel, and the results provide new insights, despite demonstrations with a small dataset (MNIST). The authors’ rebuttal has successfully addressed the major concerns of reviewers. Therefore, I recommend acceptance of this submission.

**Audience:**

Yes

**Audience Explanation:**

Of broad interest.

**Claims And Evidence:**

Yes

**Claims Explanation:**

The claims are supported by the empirical results.